# Cooperative role of PACT and ADAR1 in preventing aberrant PKR activation by self-derived double-stranded RNA

Lavanya Manjunath [1,2], Gisselle Santiago[1,2], Pedro Ortega [1,2], Ambrocio Sanchez [1,2], Sunwoo Oh [1,2], Alexander Garcia[1,2], Junyi Li [1,2], Dana Duong [1,2], Elodie Bournique [1,2], Alexis Bouin[2,3], Bert L. Semler [2,3], Dheva Setiaputra [4] & Rémi Buisson [1,2,5] ✉

Double-stranded RNAs (dsRNAs) produced during viral infections are recognized by the innate immune sensor protein kinase R (PKR), triggering a host translation shutoff that inhibits viral replication and propagation. Given the harmful effects of uncontrolled PKR activation, cells must tightly regulate PKR to ensure that its activation occurs only in response to viral infections, not endogenous dsRNAs. Here, we use CRISPR-Translate, a FACS-based genome-wide CRISPR-Cas9 knockout screening method that exploits translation levels as a readout and identifies PACT as a key inhibitor of PKR during viral infection. We find that PACT-deficient cells hyperactivate PKR in response to different RNA viruses, raising the question of why cells need to limit PKR activity. Our results demonstrate that PACT cooperates with ADAR1 to suppress PKR activation from self-dsRNAs in uninfected cells. The simultaneous deletion of PACT and ADAR1 results in synthetic lethality, which can be fully rescued in PKR-deficient cells. We propose that both PACT and ADAR1 act as essential barriers against PKR, creating a threshold of tolerable levels to endogenous dsRNA in cells without activating PKR-mediated translation shutdown and cell death.

The innate immune system is the first line of defense against pathogens such as viruses or bacteria after they gain entry into cells[1,2]. The initial phase of the innate immune response relies on the host cell's ability to detect characteristic molecular patterns of pathogens that are absent in the host[3–5]. Virus-associated molecules such as glycoproteins, genomic DNA, genomic RNA, or double-stranded RNA (dsRNA) are recognized by pattern-recognition receptors (PRRs) expressed in the host cells[3–5]. Upon recognition, the PRRs trigger different signaling pathways that initiate various responses to inhibit viral replication and prevent subsequent rounds of infections[5]. The molecular structure of DNA and RNA is universal across organisms, making it challenging for host cells to differentiate self from non-self-DNA or RNA[6,7]. In eukaryotic cells, DNA is sequestered within the nucleus, allowing for the evolution of detection mechanisms that specifically target cytoplasmic viral DNA. In contrast, host RNAs are present in both the nucleus and cytoplasm. Therefore, PRRs rely on recognizing specific RNA structural motifs such as secondary structures (e.g., dsRNA), the length of the RNA, or RNA modifications found predominantly in viral genomes or produced during viral replication in host cells[6,8]. However, dsRNA structures can also be found in human

[1]Department of Biological Chemistry, School of Medicine, University of California Irvine, Irvine, California, USA. [2]Center for Virus Research, University of California Irvine, Irvine, California, USA. [3]Department of Microbiology & Molecular Genetics, School of Medicine, University of California Irvine, Irvine, California, USA. [4]Department of Molecular Biology and Biochemistry, Simon Fraser University, Burnaby, British Columbia, Canada. [5]Department of Pharmaceutical Sciences, School of Pharmacy & Pharmaceutical Sciences, University of California Irvine, Irvine, California, USA. ✉e-mail: rbuisson@uci.edu

cells, primarily arising from the transcription of inverted repeat Alu elements[9,10]. Hence, it is crucial for cells to harbor PRR mechanisms that can either tolerate or distinguish self-dsRNA from non-self-dsRNA. An imbalance between immune activity and self-tolerance can lead to immune disorders and increase susceptibility to infectious diseases, underscoring the importance of further characterizing the mechanisms that prevent self-dsRNA from triggering an innate immune response[6,7].

The protein kinase R (PKR) pathway is one of the innate immune signaling systems activated in response to cytoplasmic dsRNAs produced during viral replication and transcription, leading to host cell translation shutdown to restrict viral protein synthesis[11,12]. PKR is a serine-threonine kinase of 551 amino acids (62 kDa) organized in two domains: an N-terminal double-stranded RNA binding region composed of two double-stranded RNA binding domains (dsRBD, also referred to as dsRBM or DRBD) and a C-terminal kinase domain[12,13]. Upon viral dsRNA binding to the PKR N-terminal RNA-binding domain, PKR forms a homodimer, resulting in its autophosphorylation at multiple serine and threonine sites, which fully activates its catalytic function[12,14]. PKR activation requires RNA molecules of at least 30 base pairs in length to allow the binding of two PKR monomers[15]. Once activated, PKR phosphorylates eIF2α at serine 51, inhibiting its function in recruiting the initiator methionyl-tRNA to the ribosome during translation initiation[12,16]. Therefore, eIF2α inhibition disrupts canonical AUG-dependent translation initiation of both host and viral mRNAs, preventing viral replication[13].

The precise regulation of PKR within cells is crucial to prevent self-dsRNA, such as inverted repeat Alu RNAs, from triggering translation shutdown while still allowing its activation during actual viral infections[17–19]. In uninfected cells, Adenosine deaminase 1 (ADAR1) binds and modifies self-dsRNA, preventing the aberrant activation of PKR[19]. ADAR1 edits RNA by converting adenosine (A) to inosine (I) in dsRNA, generating multiple mismatches and stopping PKR binding and activation by duplex RNA[19,20]. However, multiple studies have shown that ADAR1's ability to prevent PKR activation does not strictly depend on its deaminase activity[19,21–23]. Instead, ADAR1's RNA-binding domains appear to play a critical role, indicating a possible mechanism of competition for dsRNA binding that limits PKR's access to RNA. Moreover, ADAR1 has been shown to interact with the PKR kinase domain, preventing its activation[22], further suggesting that ADAR1's role in regulating PKR goes beyond editing RNAs to prevent PKR binding to duplex RNAs.

Aberrant PKR activation in cells has been implicated in the pathogenesis of myotonic dystrophy and neurodegenerative diseases such as Alzheimer's, Parkinson's, Huntington's disease, and amyotrophic lateral sclerosis (ALS)[24–26]. The mechanisms of PKR activation in these diseases are still being investigated, underscoring the importance of tightly regulating PKR in the absence of viral infections. The dual requirement for cells to suppress aberrant PKR activation by self-dsRNA but allow its activation during viral infection implies the presence of a fine-tuned regulatory mechanism. However, it remains uncertain whether ADAR1 is the sole factor preventing PKR activation from endogenous dsRNAs.

In this study, we use CRISPR-Translate, a FACS-based genome-wide CRISPR library screening method developed by our laboratory[27], to identify factors regulating PKR. One such factor that we identified is PACT, which limits PKR activation in cells infected with different types of RNA viruses. Moreover, we discovered that the role of PACT in suppressing PKR activation is critical for preventing self-dsRNA from triggering aberrant activation of PKR in uninfected cells. We show that PACT works together with ADAR1 in preventing PKR hyperactivation by endogenous RNAs. Depletion of both PACT and ADAR1 leads to synthetic lethality, demonstrating the presence of two layers of protection against uncontrolled activation of PKR-mediated translation arrest and cell death in cells by endogenous dsRNAs.

## Results

### CRISPR-Translate: A CRISPR-Cas9 screening method to identify factors regulating translation during viral infection

To reveal key factors that regulate PKR in response to viral infections, we applied the CRISPR Translate method, a FACS-based CRISPR-Cas9 screening strategy developed in our laboratory[27], which exploits ongoing translation levels as a readout to identify factors that either enhance or inhibit translation in response to specific stress conditions (Fig. 1A). In this study, we selected Sendai virus (SeV), a single-stranded, negative-sense RNA virus known to activate PKR in infected cells, without triggering the OAS3-RNase L innate immune pathway that also induces translation arrest in response to cytoplasmic dsRNA[28,29]. Therefore, translation arrest following SeV infection is strictly dependent on PKR activity[29]. To perform CRISPR-Translate, we first transduced U2OS cells with the genome-wide Brunello CRISPR library containing 76,441 gRNAs targeting 19,114 genes at a multiplicity of infection (MOI) of ~ 0.3, and uninfected cells were removed using puromycin selection (Fig. 1A, steps 1-2). The transduced cells harboring the library covering 99.85% of the Brunello library gRNAs were then infected with SeV for 15 h, corresponding to the time when SeV began to activate PKR (Fig. 1A, step 3 and Supplementary Fig. 1A). Importantly, for the purpose of the screen, we selected U2OS cells that showed a higher infection efficiency with SeV and lentivirus compared to the A549 cell line. During the last 30 min of the infection with SeV, cells were treated with azidohomoalanine (AHA)[30], an analog of methionine that is incorporated into newly synthesized polypeptide chains as a marker of translation levels (Fig. 1A, step 4). AHA was then labeled with a 488-tagged alkyne probe using Click-iT reaction (Fig. 1A, step 4). Finally, cells were FACS-sorted into two groups: those with positive 488 fluorescence signals, indicating active translation, and those negative for 488 signals, reflecting translation arrest (Fig. 1A, step 5). Genomic DNA was subsequently extracted from both populations, followed by PCR amplification of gRNA sequences for deep sequencing and quantification to identify genes essential for regulating translation arrest in response to SeV infection (Fig. 1A, step 6).

### PACT prevents translation shutdown in response to RNA viral infections

Following deep sequencing of the gRNAs recovered in the 488-positive and -negative cell populations, we used MAGeCK (Model-based Analysis of Genome-wide CRISPR-Cas9 Knockout)[31] to identify gRNAs specifically enriched in each of those two populations. First, we looked at gRNAs significantly enriched in the cell population without translation arrest (488-positive cells). As expected, *EIF2AK2* (PKR) was a top target that promoted translation arrest following SeV infection (Fig. 1B and Supplementary Data 1), validating the CRISPR-Translate screening strategy as an effective method to discover factors regulating translation in response to viral infections. We then monitored gRNAs that promote translation shutdown and, therefore, enriched in the 488-negative cell population. The top target gene identified in this cell population was *MARS*. MARS (Methionyl-tRNA synthetase) is an enzyme essential for catalyzing the attachment of methionine (and potentially AHA) to its corresponding transfer tRNA[32]. Therefore, the absence of MARS does not necessarily reflect its involvement in translation regulation after SeV infection, but rather the lack of AHA-tRNA formation by the cells for labeling nascent proteins. The second top target gene found to enhance translation arrest was *PRKRA*, encoding for the protein PACT. PACT is a dsRNA-binding protein of 313 amino acids (34 kDa) composed of three dsRNA binding domains (dsRBD1-3), and was initially identified as an interactor and activator of PKR in vitro[33–36]. However, PACT's biological function in cells remains unclear and controversial[8].

To investigate the role of PACT in translation regulation following SeV infections, we generated PACT knockout (KO) in U2OS and A549 cells (Supplementary Fig. 1B, C). To validate the CRISPR-Translate findings, we infected both wild-type (WT) or PACT KO cells with SeV

and monitored translation arrest using puromycin. Puromycin, a structural analog of aminoacyl-tRNAs, is incorporated into nascent peptides by translating ribosomes, serving as a marker for translation activity[37]. We opted for puromycin over AHA to avoid potential off-target effects that could interfere with AHA incorporation in cells. PACT KO in both U2OS and A549 cells showed a strong increase in the percentage of puromycin-negative cells following SeV infection (Fig. 1C, D and Supplementary Fig. 1D). In contrast, PACT KO cells

complemented with PACT WT fully restored translation in those cells (puromycin-positive) (Fig. 1E and Supplementary Fig. 1E). These results demonstrate that PACT is critical to limit translation shutdown in response to SeV infection. Importantly, we did not observe a difference between WT and PACT KO cells in SeV infection rate when monitoring the percentage of SeV-positive cells using fluorescent in situ hybridization (RNA FISH) to detect SeV RNA genomes in infected cells (Supplementary Fig. 1F, G)[38], ruling out any difference in infectivity caused

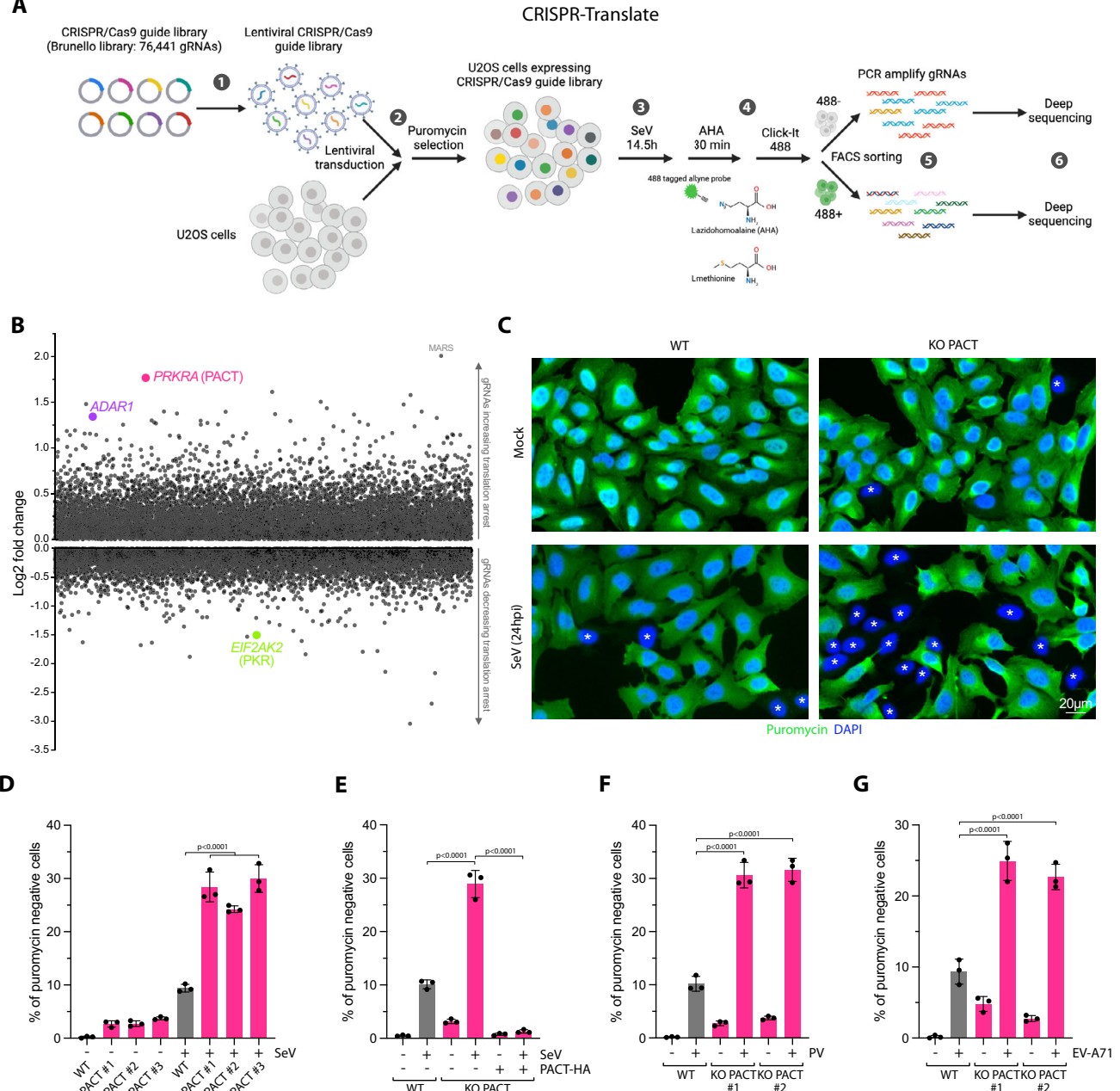

**Fig. 1 | CRISPR-translate identifies PACT as a suppressor of translation arrest during RNA viral infection. A** Schematic representation of the CRISPR-Translate screening approach. Created in BioRender. Buisson, R. (2025) https://BioRender.com/v58p847. **B** Scatter plot showing the log fold change of genes enriched in the 488-negative FACS-sorted cell population, analyzed using the MaGeCK computational tool. Each dot represents a unique gene. **C** Representative immunofluorescence for puromycin in U2OS WT and PACT KO cells infected with SeV (MOI = 1, 24hpi). Cells undergoing translation arrest are marked with an asterisk. **D** Quantification of puromycin-negative cells (%) in U2OS WT or PACT KO cells

infected with SeV (MOI = 1, 24hpi). Mean values ± SD (Number of biological replicates, $n$ = 3). **E** Quantification of puromycin-negative cells (%) in U2OS WT or PACT KO cells infected with SeV (MOI = 1, 24hpi). When indicated, PACT-HA was expressed. Mean values ± SD (Number of biological replicates, $n$ = 3). **F**, **G** Quantification of puromycin-negative cells (%) in the indicated cell lines infected with PV (MOI = 1, 16hpi) (**F**), or EV-A71 (MOI = 5, 24hpi) (**G**). Mean values ± SD (Number of biological replicates, $n$ = 3). All $P$-values were calculated by one-way ANOVA. Source data are provided as a Source Data file.

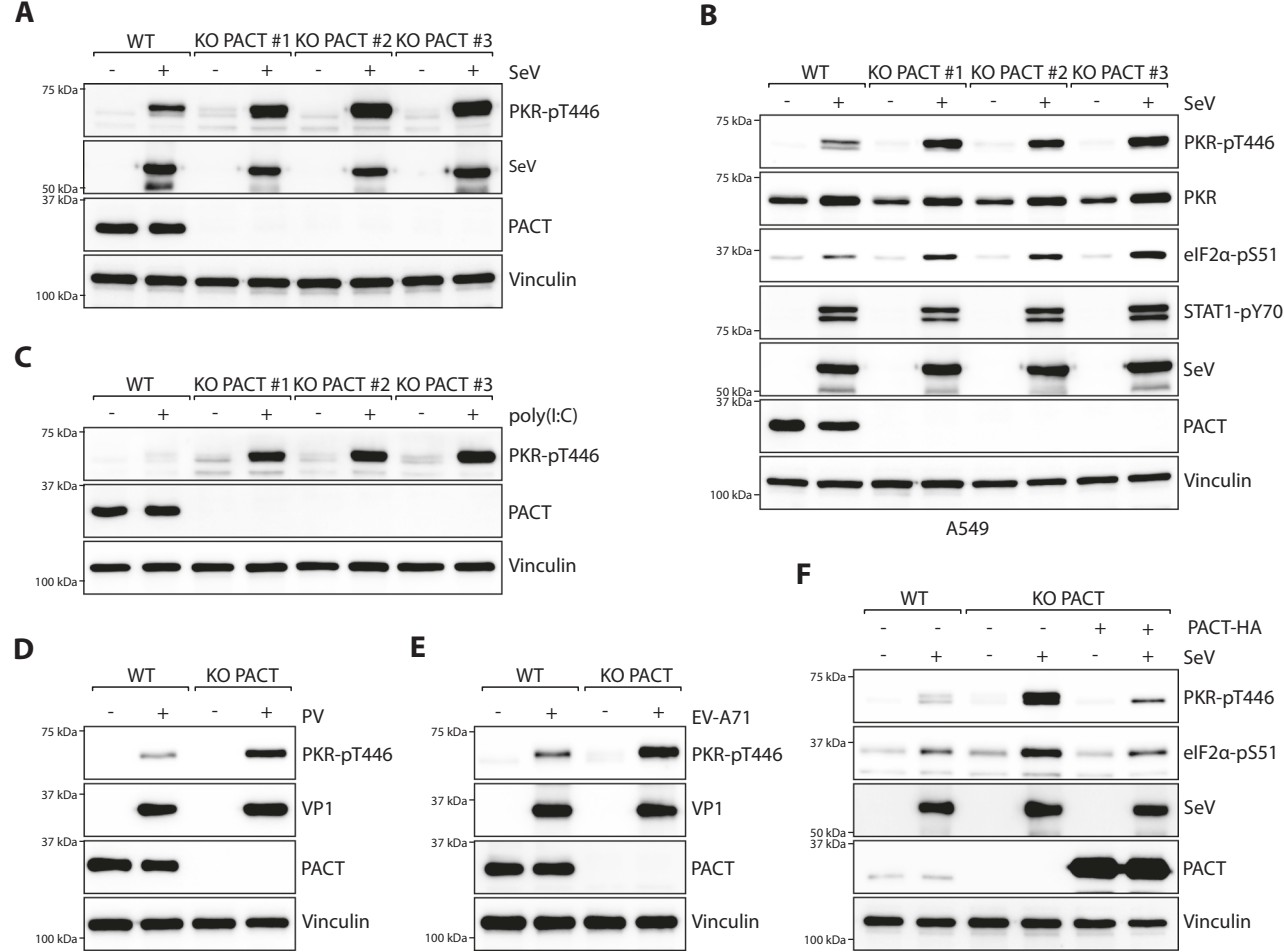

**Fig. 2 | PACT inhibits PKR phosphorylation during RNA viral infection.**
**A**, **B** U2OS (**A**) or A549 (**B**) WT and PACT KO cells were infected with SeV (MOI = 1, 24hpi). The levels of PKR phosphorylation (PKR-pT446) and other indicated proteins were monitored by western blot. **C** U2OS WT and PACT KO cells were transfected with 200 ng/mL of poly(I:C) for 6 h. The levels of PKR-pT446, PACT, and Vinculin were monitored by western blot. **D**, **E** U2OS WT and PACT KO cells were infected with PV (MOI = 1, 20hpi) (**D**) or EV-A71 (MOI = 5, 24hpi) (**E**). The levels of the indicated proteins were monitored by western blot. **F** The levels of PKR-pT446, eIF2α-pS51, SeV, PACT, and Vinculin were analyzed in the indicated U2OS cell lines by western blot after SeV infection (MOI = 1) at 24hpi. PACT-HA was expressed when indicated. Source data are provided as a Source Data file.

by the absence of PACT in cells. We then asked whether PACT prevents translation shutdown was specific to SeV infection or if it also occurs with other RNA virus infections. We infected cells with poliovirus (PV) or enterovirus A-71 (EV-A71), two positive-sense single-strand RNA viruses. Similar to SeV infection, PACT KO cells infected with PV or EV-A71 showed higher levels of puromycin-negative cells than WT cells (Fig. 1F, G), further confirming the role of PACT in limiting translation arrest. Taken together, these results demonstrate that PACT functions as a global suppressor of translation arrest during RNA viral infection.

**PACT suppresses PKR-mediated translation shutdown and stress granule formation**
To determine how PACT limits translation shutdown in response to viral infection, we examined whether PACT regulates PKR-mediated translation arrest. In the absence of PACT, PKR was strongly activated in both U2OS and A549 cells infected with SeV (Fig. 2A, B). Similarly, eIF2α phosphorylation increased in PACT KO cells compared to WT cells (Fig. 2B). However, the levels of STAT1 phosphorylation, as a marker of interferon (IFN) response known to be triggered in response to SeV infection[29], were not impacted by the absence of PACT in cells (Fig. 2B), further suggesting that PACT specifically regulates the PKR pathway upon viral infection. We then asked whether PKR inhibition by PACT is a specific response to SeV infection or a general mechanism that extends

to other RNA viruses. We first transfected cells with poly(I:C), a synthetic analog of dsRNA that is structurally similar to dsRNA present during viral infections and is used as a surrogate for viral dsRNA. As previously shown, poly(I:C) transfection triggers PKR phosphorylation (Fig. 2C). However, PACT KO cells exhibit significantly stronger activation of PKR compared to WT cells upon poly(I:C) transfection (Fig. 2C). Likewise, infection with single-stranded RNA viruses such as PV, EV-A71, and Sindbis virus (SINV) activates PKR, which is further amplified without PACT (Fig. 2D, E and Supplementary Fig. 2A). Importantly, PACT KO cells complemented with PACT WT completely suppressed PKR phosphorylation levels after SeV infection or poly(I:C) transfection (Fig. 2F and Supplementary Fig. 2B). Together, these results demonstrate that PACT limits PKR activation in response to RNA viral infection.

PKR activation not only promotes translation arrest in infected cells but also triggers the formation of stress granules[39,40]. Stress granules are dynamic, non-membrane-bound structures that form in the cytoplasm of eukaryotic cells under stressful conditions, such as viral infections, heat shock, or nutrient deprivation. They contain untranslated mRNAs, RNA-binding proteins, translation initiation factors, and other regulatory proteins[40–42]. These cellular structures are highly dynamic and dissolve once the stress subsides, allowing translation to resume[40–42]. To quantify the level of stress granules in SeV-infected cells, we monitored G3BP1, a core constituent of stress

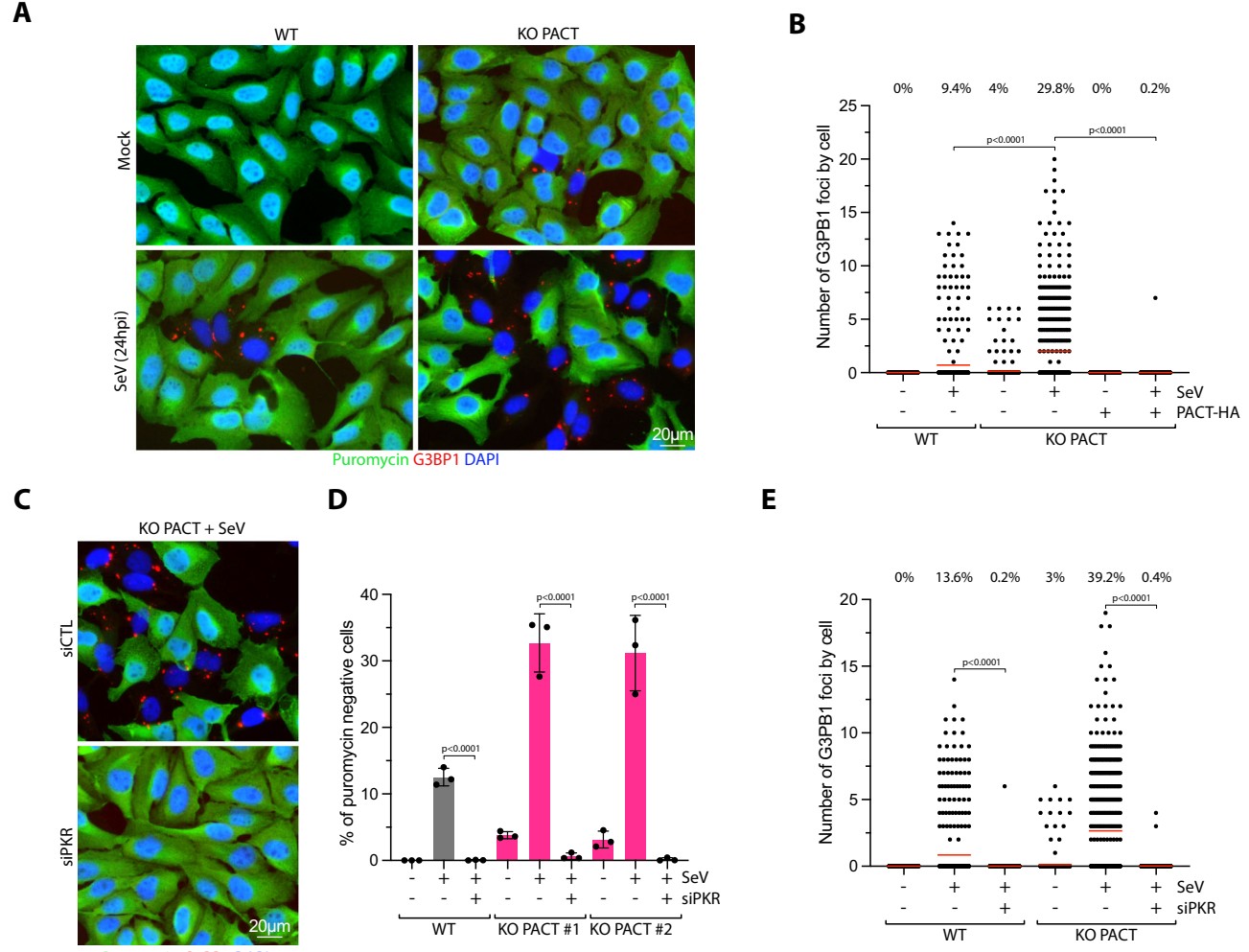

**Fig. 3 | PACT suppresses PKR-mediated translation arrest and stress granule formation during RNA viral infection. A** U2OS WT and PACT KO cells infected with SeV (MOI = 1, 24hpi) were treated with puromycin 15 min before fixation. G3BP1 and puromycin were monitored by immunofluorescence. **B** Quantification of the number of G3BP1 foci in individual cell in indicated cell lines infected with SeV (MOI = 1, 24hpi). Red lines indicate the mean (Number of cells, *n* = 500). Top; percentage of cells with G3BP1 foci. **C** U2OS PACT KO cells were knocked down with siRNA control (siCTL) or against PKR (siPKR) for 40 h and infected with SeV

(MOI = 1, 24hpi). The levels of G3BP1 and puromycin were monitored by immunofluorescence. **D** Quantification of the percentage puromycin-negative cells of the experiment shown in Fig. 3C. Mean values ± SD (Number of biological replicates, *n* = 3). **E** Quantification of the number of G3BP1 foci in individual cells of the experiment described in Fig. 3C. Red lines indicate the mean (Number of cells, *n* = 500). Top; percentage of cells with G3BP1 foci. All *P*-values were calculated by one-way ANOVA. Source data are provided as a Source Data file.

granules[40,43]. Following SeV infection, PACT KO cells strongly increased the percentage of cells with stress granules and the number of stress granules within cells compared to WT cells (Fig. 3A, B). Cells positive for stress granules were also puromycin-negative (Fig. 3A), confirming the tight link between translation levels and stress granule formation[44]. While basal levels of stress granules were detected in PACT KO cells prior to infection (Fig. 3A, B), stress granule formation was strongly enhanced following PACT depletion when quantified specifically in SeV-infected cells (Supplementary Fig. 2C, D). Moreover, stress granules were absent in PACT KO cells infected with SeV and complemented with PACT WT (Fig. 3B). In addition, PKR knockdown in WT or PACT KO cells completely suppressed puromycin-negative cell levels and stress granule formation following SeV infection (Fig. 3C–E). Of note, we did not examine stress granule formation, puromycin incorporation, or eIF2α phosphorylation levels after poly(I:C) transfection, as poly(I:C) can activate RNase L, which modulates these processes independently of PKR[45–47]. Collectively, these data demonstrate that PACT limits PKR-mediated translation arrest and stress granule formation in response to RNA viral infections.

## Structural modeling of PACT on dsRNA

To investigate how PACT influences PKR activity triggered by dsRNA, we carried out AlphaFold 3 modeling[48] of the protein-RNA interaction between PACT and a 30 bp dsRNA sequence previously shown to stimulate PKR[15]. We found that PACT is predicted to bind dsRNA either as a monomer or dimer, with direct interactions occurring between the dsRBD1 and dsRBD2 of PACT and the RNA duplex, whereas the dimerization of the two PACT molecules is predicted to occur through interactions between the dsRBD3 domains (Fig. 4A, B, Supplementary Fig. 3A, B, and Supplementary Movie 1). These results were consistent with previous biochemical characterizations of PACT[35,49–51]. The folding of three dsRBD domains of PACT is predicted with high confidence (pLDDT > 70) (Supplementary Fig. 3C), and the superposition of the five models generated by AlphaFold 3 demonstrated consistent protein folding of PACT dsRBDs (Supplementary Fig. 3D, E). Moreover, both PACT dsRBD1 and dsRBD2 are predicted to engage dsRNA analogously to other proteins containing dsRBDs, including ADAR1, ADAR2, and TRBP (Fig. 4C)[52–54], highlighting the validity of this modeling approach. Each PACT monomer's dsRBD1 and dsRBD2 is

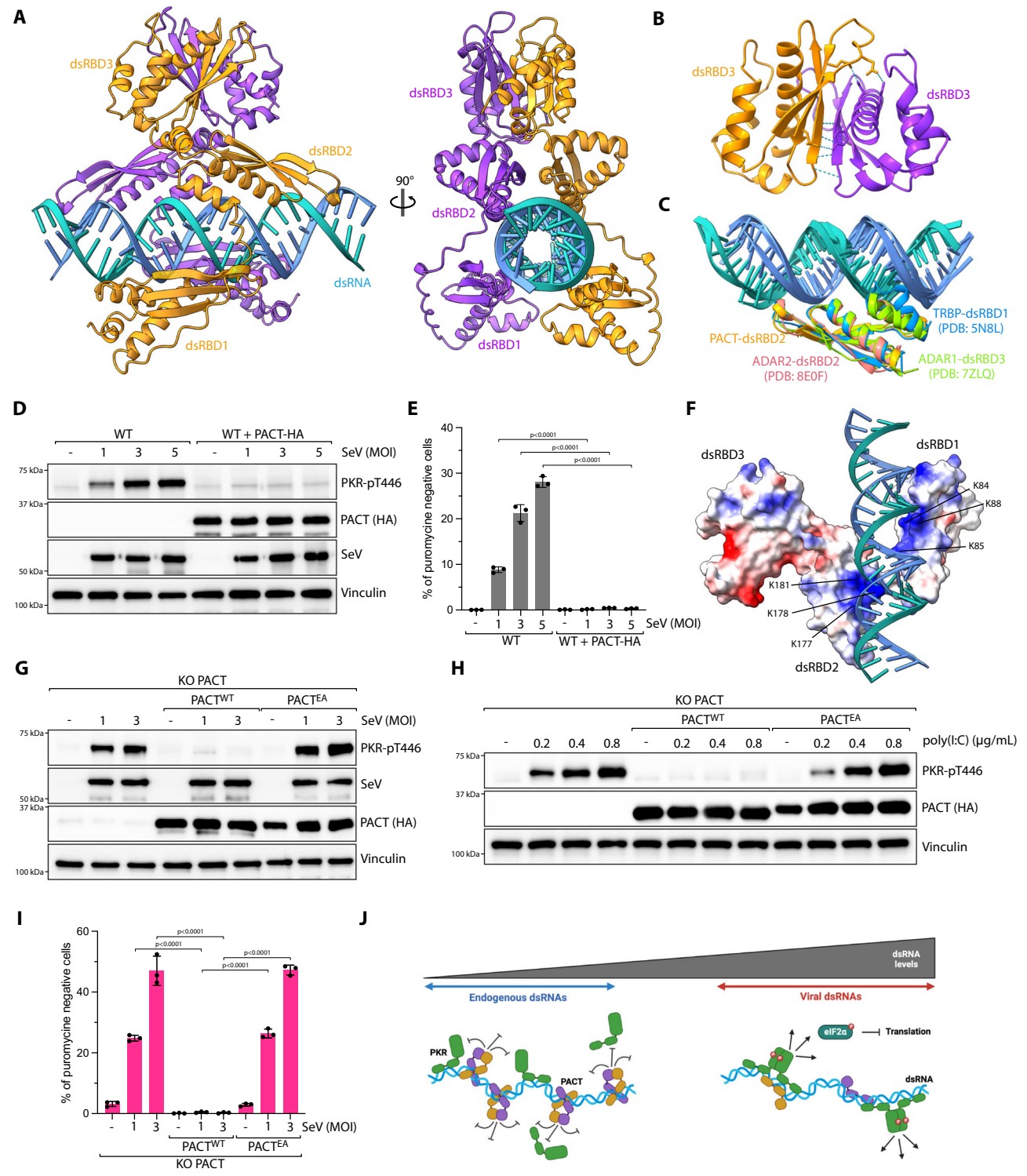

predicted to interact with opposite sides of the dsRNA, forming an X-shaped-like structure that brings both dsRBD3 domains into close proximity, facilitating dimerization by forming an extended β-sheet structure (Fig. 4A, B). PACT dsRBD3 dimerization is predicted with high confidence, with an average predicted aligned error (PAE) of 7.7 Å between the two domains (Supplementary Fig. 3F). A separate Alpha-Fold 3 prediction of two PACT dsRBD3 domains supported this model, with an iPTM score of 0.6 and mean interface PAE of 5.9 Å (Supplementary Fig. 3G). Notably, the deletion of dsRBD3 was shown not to impair PACT's RNA binding ability and may even slightly enhance its interaction with dsRNA[35,49,50]. This suggests that the conformational

shift between monomeric and dimeric PACT might not significantly impact PACT's stability on RNA. Instead, PACT dimerization could potentially facilitate the formation of a structural barrier, blocking access to both sides of the dsRNA and thereby preventing PKR activation.

## PACT inhibits PKR primarily through its RNA-binding activity

If PACT binds to dsRNA and inhibits PKR activation, then the expression levels of PACT would directly influence the cell's capacity to activate PKR in response to different dsRNA levels. This would establish a threshold for the maximum amount of dsRNA that PACT can shield from

**Fig. 4 | Structural modeling of PACT on dsRNA. A** Top-ranked model predicted by AlphaFold 3 of PACT dimer (residues 21 to 313) binding to dsRNA (blue). Each PACT monomer was distinctly color-coded in either orange or purple. Amino acids 1 to 20 of PACT, which form an unfolded polypeptide chain, were not depicted. **B** Close-up view of the interaction between β-sheets of the dsRBD3 of each PACT monomer (residues 215–311). H-bonding or ionic bonds between both PACT-dsRBD3 monomers are indicated with blue dashed lines. **C** Top-ranked model predicted by AlphaFold 3 of PACT dsRBD2 superimposed to known structures of ADAR1-dsRBD3 (PDB: 7ZLQ), ADAR2dsRBD2 (PDB: 8E0F), and TRBP-dsRBD1 (PDB: 5N8L). **D** U2OS WT or PACT-HA overexpressing cells were infected with SeV at the indicated MOI (24 hpi). The levels of PKR phosphorylation (PKR-pT446) and other indicated proteins were monitored by western blot. **E** Quantification by immunofluorescence of puromycin-negative cells (%) in U2OS WT cells infected with SeV at the indicated MOI (24hpi). When indicated, PACT-HA was expressed. Mean values ± SD (Number of biological replicates, $n = 3$). *P*-values were calculated by one-way ANOVA.

**F** Electrostatic surface potentials of PACT were color-coded: red for negatively charged amino acids, white for neutral residues, and blue for positively charged amino acids, based on the top-ranked AlphaFold 3 model of the PACT dimer (residues 21–313) bound to dsRNA. Only one PACT monomer is shown, with lysines in close proximity to dsRNA highlighted. **G, H** Indicated cell lines expressing when indicated PACT^WT or PACT^EA were infected with SeV at the indicated MOI for 24h (**G**) or transfected with the indicated concentration of poly(I:C) for 6 h (**H**). The levels of PKR phosphorylation (PKR-pT446) and other indicated proteins were monitored by western blot. **I** Quantification of puromycin-negative cells (%) in U2OS PACT KO cells infected with SeV at the indicated MOI (24hpi). When indicated, PACT^WT-HA or PACT^EA-HA were expressed. Mean values ± SD (Number of biological replicates, $n = 3$). *P*-values were calculated by one-way ANOVA. **J** Proposed model of PACT functioning as a barrier that prevents PKR binding or dimerization on dsRNA. Created in BioRender. Buisson, R. (2025) https://BioRender.com/qo988lb. Source data are provided as a Source Data file.

triggering PKR activation, allowing its activation during a viral infection. Consistent with this idea, the modeling of the PKR homodimer on dsRNA indicates that PKR can bind to one side of the dsRNA while monomeric PACT binds to the opposite side (Supplementary Fig. 4A and Supplementary Movie 2). Of note, AlphaFold 3 predicted the folding of the PKR kinase and dsRBD domains with high confidence (pLDDT > 70) that are highly consistent with previously characterized structures of the PKR kinase domain and PKR-dsRBDs (Supplementary Fig. 4B–D)[55,56]. Since our AlphaFold 3 analysis indicates that a PKR dimer and a single PACT molecule can occupy the same dsRNA motif, we speculate that if PACT levels are insufficient to protect both sides of the dsRNA, PKR can bind and become activated through dimerization. To test this hypothesis, we overexpressed PACT in wild-type cells and infected them with increasing multiplicity of infection (MOI) of SeV. PACT overexpression completely suppressed basal PKR activation and restored translation level in wild-type cells during SeV infection, even in the presence of a high level of virus (Fig. 4D, E), demonstrating that higher PACT expression is sufficient to inhibit PKR activation by raising the threshold of dsRNA that can be protected from PKR recognition.

We next asked whether PACT binding to RNA is required to suppress PKR. Our structural modeling of PACT on dsRNA revealed the presence of two basic patches (patch 1 and patch 2) that are predicted to directly interact with dsRNA (Fig. 4F). Patch1 and patch 2 are located in dsRBD1 and dsRBD2, respectively, and consist of three lysines each (patch 1: K84, K85, K88; patch 2: K177, K178, K181) that fit into the major groove of the RNA duplex (Fig. 4F). This structural organization of lysine residues interacting with dsRNA in PACT is similar to other proteins with dsRBDs (Supplementary Fig. 4E), and is known to be essential for the binding to dsRNAs[19,57,58]. We then mutated the lysines in the PACT dsRBD1 and dsRBD2 domains to alanines or to glutamate to increase negative charges and disturb the interaction with RNA (PACT^EA: K84E, K85E, K88A in patch 1; K177E, K178A, K181A in patch 2). PACT^EA expression failed to suppress PKR phosphorylation in PACT KO cells infected with SeV or transfected with poly(I:C) (Fig. 4G, H), and did not rescue translation levels upon SeV infection (Fig. 4I), strongly suggesting that PACT binding to dsRNA is essential for inhibiting PKR activation. Taken together, we propose that PACT prevents hyperactivation of PKR, either through direct competition or by creating a barrier that prevents the assembly of two PKR monomers bound on the dsRNA (Fig. 4J).

### ADAR1 depletion causes synthetic lethality in PACT-deficient cells

The inhibition of PKR by PACT during viral infection raises the fundamental question: why would cells want to limit PKR activation? In uninfected cells, the loss of PACT in both U2OS and A549 cells triggers spontaneous translation arrest and stress granule formation in a small percentage of cells in a PKR-dependent manner (Figs. 1C-D and 2A, B). Moreover, PACT KO cells showed elevated levels of PKR

phosphorylation compared to wild-type cells (Supplementary Fig. 5A). These observations indicate that PACT may have a broader role beyond viral infection, protecting cells from the aberrant activation of PKR by endogenous RNAs. Therefore, we propose that PACT acts as a buffer to limit PKR activation caused by low basal levels of dsRNA that are present in uninfected cells. However, only a small percentage of puromycin-negative and SG-positive cells were detected in uninfected PACT KO cells (Figs. 1D, 3D, E). This suggests that PACT may work with other factors to prevent PKR activation by self-RNAs. To unbiasedly identify factors that work with PACT to avoid PKR-mediated cell death by recognition of endogenous dsRNA, we conducted a genome-wide CRISPR-Cas9 sgRNA dropout screen by comparing wild-type cells with PACT KO cells. Cells were transduced with the genome-wide Brunello CRISPR library, selected with puromycin for 4 days, and subsequently cultured for 7 days before collecting the cells and purifying gRNAs for deep sequencing (Fig. 5A and Supplementary Data 1). Remarkably, we found that gRNAs targeting *ADAR1* were preferentially depleted in PACT KO cells (Fig. 5B). Consistently, ADAR1 also emerged as one of the top regulators of translation in response to SeV infection in our CRISPR-Translate screen (Fig. 1B). We first validated this result by knocking down ADAR1 using siRNAs. Both U2OS and A549 PACT KO cells depleted of ADAR1 exhibited rapid cell death within 6 days following transfection with a siRNA targeting ADAR1 (Fig. 5C–E Supplementary Fig. 5B). In contrast, neither PACT KO cells nor cells with ADAR1 knockdown alone displayed substantial cell growth defects compared to cells depleted for both PACT and ADAR1 (Supplementary Fig. 5C). However, PACT KO cells depleted of ADAR1 and complemented with PACT WT rescued cell survival and cell growth, whereas complementation with PACT^EA mutant did not (Fig. 5F, G and Supplementary Fig. 5C). Finally, we knocked down ADAR2 in PACT KO cells, which is a homolog of ADAR1, sharing conserved domains and a similar structural organization[17]. However, cells lacking ADAR2 did not show any increased cell death in the absence of PACT (Supplementary Fig. 5D–F), suggesting that the cell's dependency on PACT is specifically tied to the absence of ADAR1. Taken together, these results reveal that ADAR1 and PACT are synthetic lethal in the absence of any infections or stresses.

### PACT and ADAR1 depletion induce PKR activation

To investigate the mechanism by which PACT and ADAR1 depletion in cells triggers rapid cell death, we monitored PKR activation. In the absence of both PACT and ADAR1, PKR and its downstream target eIF2α, were highly activated, displaying high phosphorylation levels compared to wild-type cells or single depletions (Fig. 6A and Supplementary Fig. 6A, B). In addition, PACT KO cells with ADAR1 knockdown exhibited high levels of translation-arrested cells that were also positive for SGs (Fig. 6B). The PKR phosphorylation levels, translation-arrested cells, and SG-positive cells were fully restored in cells complemented with wild-type PACT but not with PACT^EA (Fig. 6C–E and

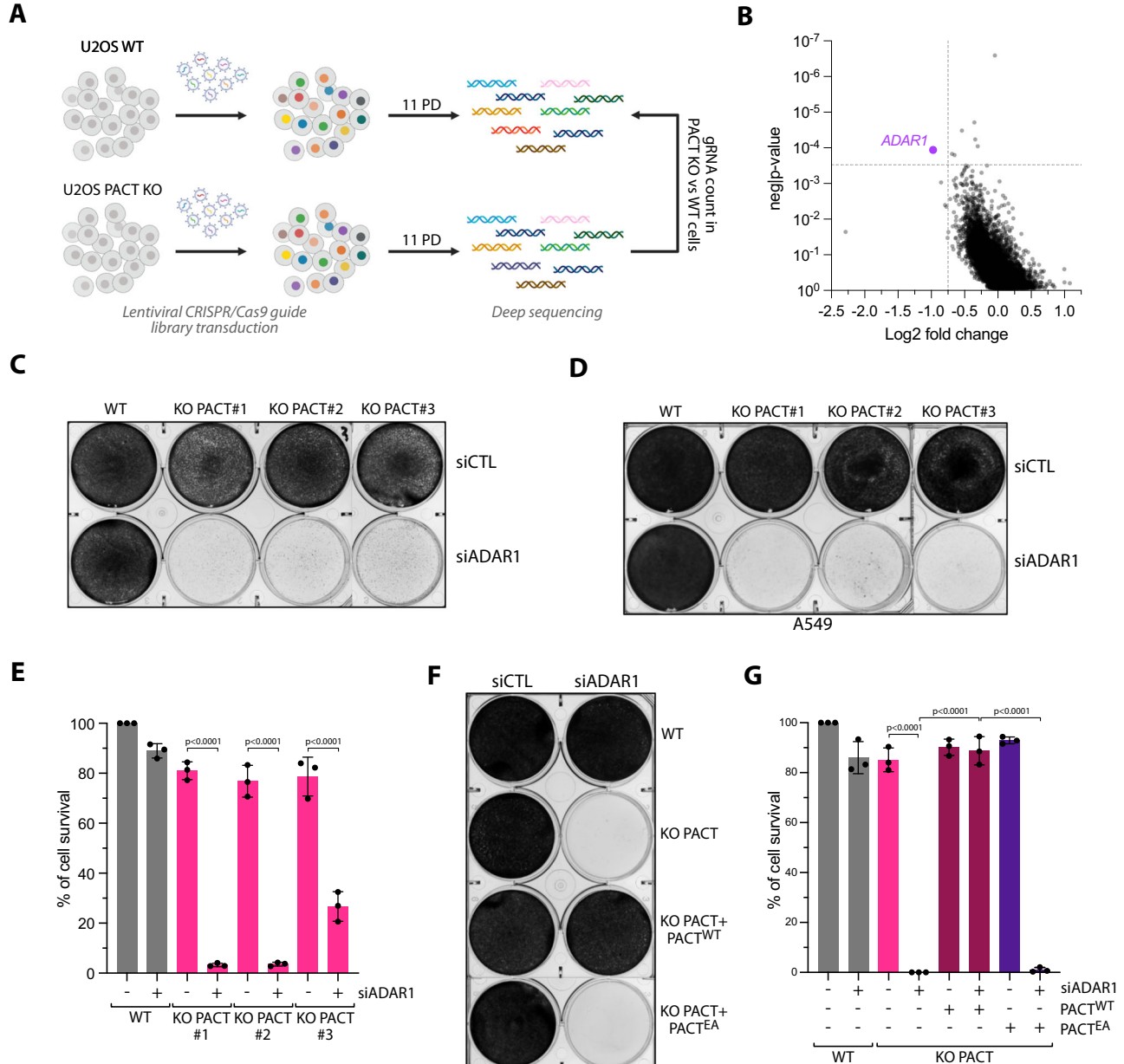

**Fig. 5 | Loss of PACT triggers synthetic lethality in ADAR1-deficient cells.**
**A** Schematic representation of CRISPR-dropout screening approach. PD; population doubling. Created in BioRender. Buisson, R. (2025) https://BioRender.com/f77e526. **B** Dot plots graph representative of the genes depleted in PACT KO cells analyzed using the MaGeCK computational tool. Each dot represents a unique gene. *ADAR1* gene was found to be significantly depleted in PACT KO cells and was highlighted in purple. **C, D** Crystal violet staining showing the viability of U2OS (**C**) or A549 (**D**) WT or PACT KO cells transfected with siRNA control (siCTL) or against ADAR1 (siADAR1). Cells were stained with Crystal violet 6 days following transfection with siRNA. **E** U2OS WT or PACT KO cells were knocked down with siCTL or siADAR1. Cell survival was quantified with Alamar blue cell viability assay 6 days following siRNA transfection. Mean values ± SD (Number of biological replicates, $n = 3$). P-values were calculated by one-way ANOVA. **F** Crystal violet staining showing viability of U2OS WT, PACT KO, PACT KO + PACT$^{WT}$, or PACT KO + PACT$^{EA}$ cells transfected with siRNA control (siCTL) or against ADAR1 (siADAR1). Cells were stained with Crystal violet 6 days following transfection with siRNA. **G** Indicated cell lines, expressing when indicated PACT$^{WT}$ or PACT$^{EA}$ were knocked down with siCTL or siADAR1. Cell survival was quantified with Alamar blue cell viability assay 6 days following siRNA transfection. Mean values ± SD (Number of biological replicates, $n = 3$). P-values were calculated by two-way ANOVA. Source data are provided as a Source Data file.

Supplementary Fig. 6C, D). We then asked whether phosphorylation sites previously identified on serine 18 near dsRBD1 or serine 167 within dsRBD2 of PACT[59,60] influence its ability to suppress PKR activity. Expression of PACT with alanine substitutions to prevent phosphorylation or aspartic acid substitutions to mimic phosphorylation at S18 or S167, also restored translation levels in PACT KO cells (Supplementary Fig. 6E), suggesting that these phosphorylation sites are not essential in regulating PACT activity and thus in preventing PKR activation.

## Overlapping roles of PACT and ADAR1 in preventing PKR activation

We then compared PKR activation resulting from SeV infection and /or ADAR1 depletion in WT and PACT KO cells. Cells knocked down for ADAR1 had similar PKR phosphorylation levels to those observed following SeV infection in PACT KO cells (Supplementary Fig. 6F). Moreover, SeV infection and ADAR1 depletion in PACT KO cells did not further increase PKR activity (Supplementary Fig. 6F). A similar result was obtained after cell transfection with an increasing concentration

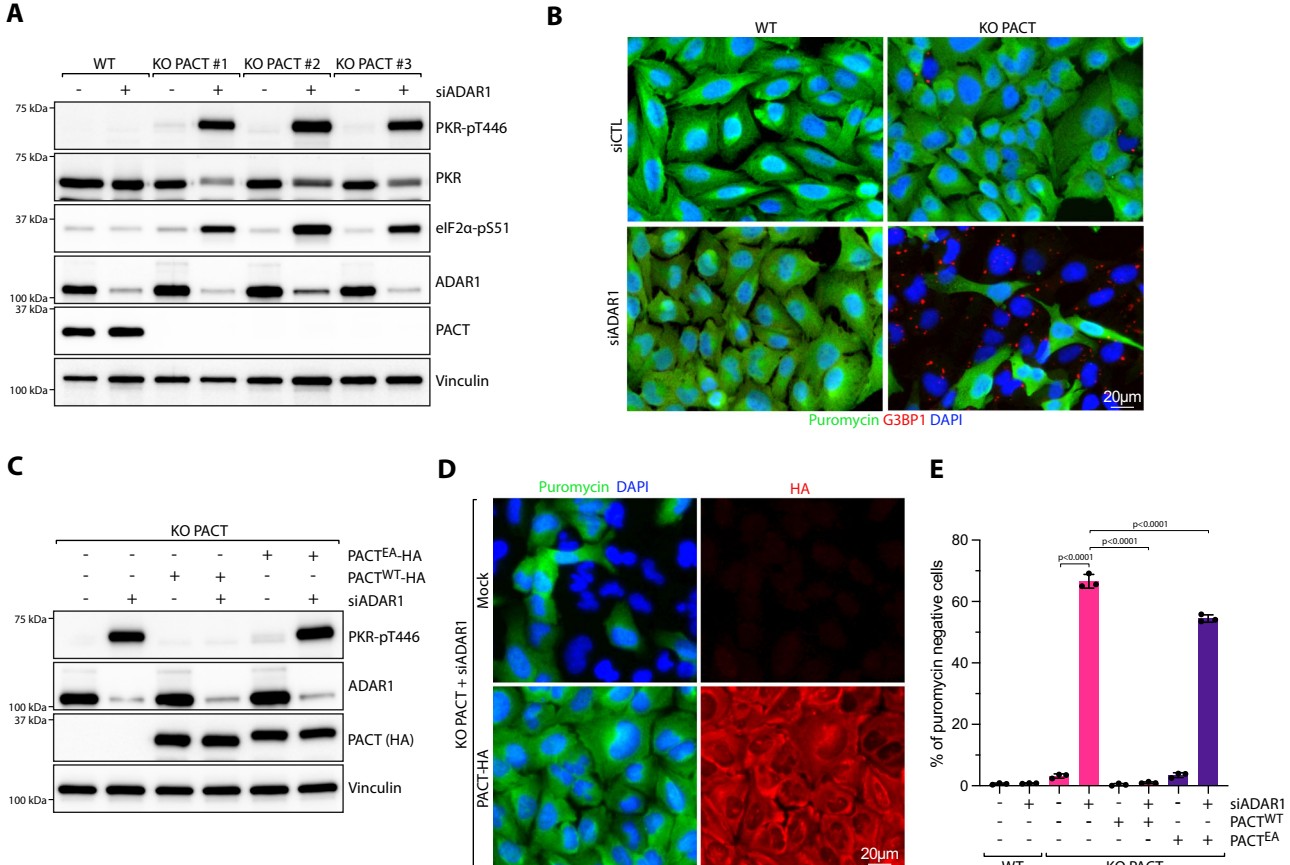

**Fig. 6 | Synergistic role of PACT and ADAR1 in suppressing PKR activation.**
**A** U2OS WT and PACT KO cells were knocked down with siRNA control (siCTL) or against ADAR1 (siADAR1) for 48 h. The levels of the indicated proteins were monitored by western blot. **B** Representative immunofluorescence for puromycin and G3BP1 in U2OS WT or PACT KO cells knocked down with siCTL or siADAR1 for 48 h. **C** The levels of PKR-pT446, ADAR1, PACT (HA), and Vinculin were analyzed by western blot in the indicated U2OS cell lines knocked down with siRNA control (siCTL) or against ADAR1 (siADAR1) for 48 h. When indicated, cells were

complemented with PACT^WT-HA or PACT^EA-HA. **D** U2OS PACT KO cells complemented with PACT-HA were knocked down with siCTL or siADAR1 for 48 h and treated with puromycin 15 min before fixation. PACT-HA and puromycin were monitored by immunofluorescence using indicated antibodies. **E** Quantification of puromycin-negative cells (%) in the indicated cell lines knocked down with siRNA control (siCTL) or against ADAR1 (siADAR1). Mean values ± SD (Number of biological replicates, $n = 3$). $P$-values were calculated by one-way ANOVA. Source data are provided as a Source Data file.

of poly(I:C), stimulating comparable levels of PKR activity to those in PACT KO cells depleted for ADAR1 (Supplementary Fig. 6G). This result further indicates that in the absence of ADAR1 and PACT, PKR activity reached a maximum level of activation in cells. Notably, the increase in phosphorylated PKR levels following ADAR1 depletion in SeV-infected cells can be reversed by overexpressing PACT (Supplementary Fig. 6H), suggesting that ADAR1 and PACT have redundant roles in preventing PKR activation triggered by viral dsRNA. Next, we complemented PACT KO cells, which were either infected with SeV or transfected with poly(I:C), by expressing wild-type PACT at different levels. To achieve this, we established two stable cell lines using distinct lentiviral vector systems: pLenti, which drives high PACT expression, and pInducer20, which induces low PACT expression levels upon doxycycline (DOX) treatment. While high PACT expression levels completely suppress PKR activation following SeV infection or poly(I:C) transfection, lower expression levels of PACT lead to partial suppression (Supplementary Fig. 6I). These findings demonstrate that differential PACT expression results in a dose-dependent response of PKR activation.

The *ADAR1* gene encodes two isoforms in cells: ADAR1-p110, which is highly expressed and located in the nucleus, and ADAR1-p150, which is expressed at lower levels and present in the cytoplasm[17]. We knocked out ADAR1-p150, reasoning that since ADAR1-p150 is located in the cytoplasm alongside PACT and PKR, unlike ADAR1-p110, its

removal would not impact the other essential functions of ADAR1 in the nucleus[17]. Depletion of PACT with a siRNA in ADAR1-p150 KO cells was sufficient to trigger hyperactivation of PKR (Fig. 7A), further confirming that either PACT or ADAR1 is necessary to prevent PKR activation by self-RNA in the absence of stresses. Next, we knocked down ADAR1 in PACT KO using a specific siRNA targeting the 3'UTR of *ADAR1*. Ectopic expression of ADAR1-p150 was sufficient to suppress both PKR activation and translation arrest in those cells (Fig. 7B, C). In addition, PACT KO cells depleted of ADAR1 and complemented with ADAR1-p150 rescued cell survival (Fig. 7D, E). We then asked whether ADAR1 deaminase activity was required. Expression of the ADAR1 catalytically-inactive mutant (ADAR1-p150-E912A)[19,21,23], also restored translation levels in PACT KO cells depleted for ADAR1 (Fig. 7C). This result is consistent with previous studies suggesting that ADAR1 deaminase activity is not essential to prevent PKR activation from endogenous RNAs[19,22,23] and suggests that ADAR1 works together with PACT by binding dsRNA to prevent PKR activation. To further support this model, we expressed ADAR1-p150 in PACT KO cells infected with increasing MOI of SeV or levels of poly(I:C). ADAR1-p150 expression was sufficient to fully suppress PKR phosphorylation levels (Fig. 7F and Supplementary Fig. 7A), demonstrating that PACT and ADAR1 have redundant functions in preventing PKR activation in response to RNA viral infection. We next monitored and quantified PKR localization at dsRNA-induced foci (dRIFs), which are known to colocalize with both

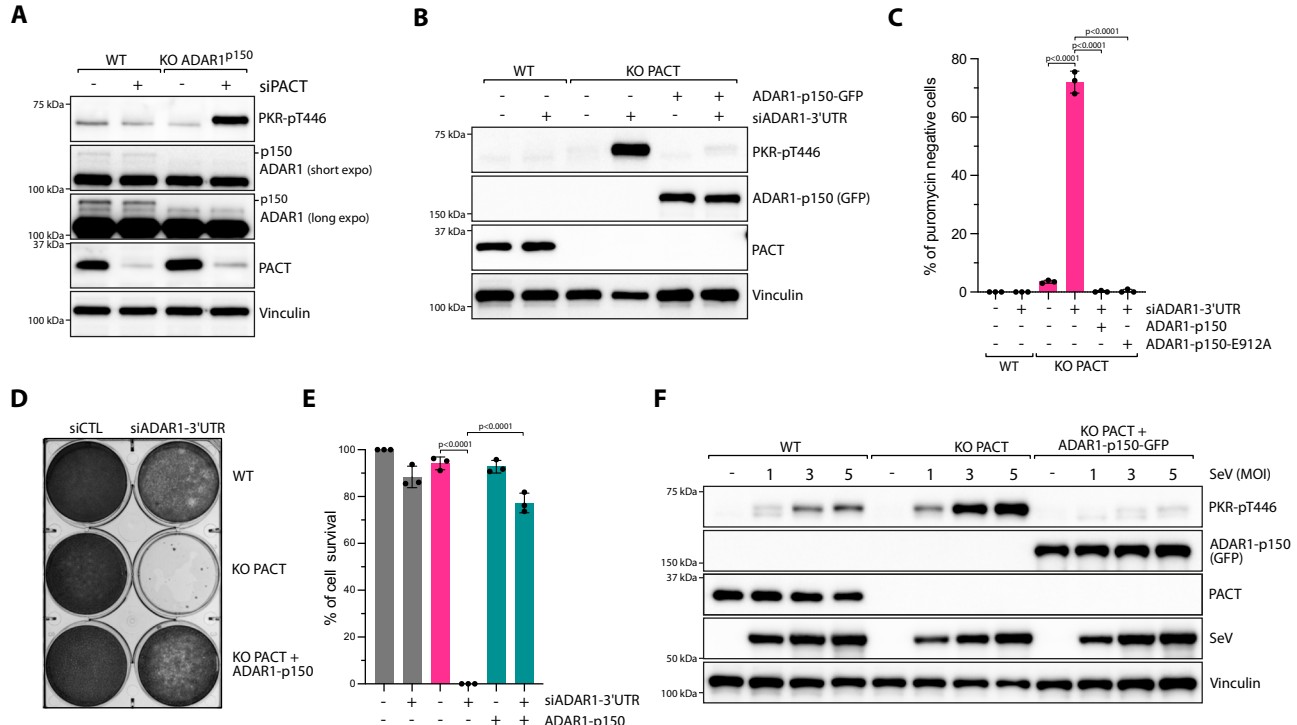

**Fig. 7 | ADAR1-p150 expression is sufficient to suppress PKR activation. A** U2OS WT or ADAR1-p150 KO cells were knocked down with siRNA control (siCTL) or against PACT (siPACT) for 48 h. The levels of the indicated proteins were monitored by western blot. **B** U2OS WT and PACT KO cells were knocked down with siRNA control (siCTL) or against 3'UTR of ADAR1 (siADAR1) for 48 h. When indicated, ADAR1-p150-GFP was overexpressed. The levels of PKR phosphorylation (PKR-pT446) and other indicated proteins were monitored by western blot.
**C** Quantification of puromycin-negative cells (%) in the indicated cell lines knocked down with siRNA control (siCTL) or against the 3'UTR of ADAR1 (siADAR1) for 48 h. When indicated, ADAR1-p150 or ADAR1-p150-E912A were transfected 16 h following siRNA transfection. Mean values ± SD (Number of biological replicates, $n = 3$).

*P*-values were calculated by two-way ANOVA. **D** Crystal violet staining showing the viability of U2OS WT, PACT KO, PACT KO + ADAR1-p150 cells transfected with siRNA control (siCTL) or against ADAR1 (siADAR1-3'UTR). Cells were stained with Crystal violet 6 days following transfection with siRNA. **E** Indicated cell lines expressing ADAR1-p150 when indicated, were transfected with siCTL or siADAR1-3'UTR. Cell survival was quantified with Alamar blue cell viability assay 6 days following siRNA transfection. Mean values ± SD (Number of biological replicates, $n = 3$). *P*-values were calculated by two-way ANOVA. **F** U2OS WT, PACT KO, or PACT KO + ADRA1-p150-GFP cells were infected with SeV at the indicated MOI. The levels of PKR phosphorylation (PKR-pT446) and other indicated proteins were monitored by western blot at 24 hpi. Source data are provided as a Source Data file.

---

ADAR1 and PACT[61,62]. Cells transfected with poly(I:C)-FITC formed large foci of PKR, PKR-pT446, and PACT associated with dsRNA (Supplementary Fig. 7B, C). However, overexpression of either PACT or ADAR1-p150 significantly reduced both the number of PKR foci and the percentage of cells with PKR foci in poly(I:C)-transfected cells (Supplementary Fig. 7D). This further suggests that PACT and ADAR1 play overlapping roles in preventing PKR association on dsRNAs in cells.

### PACT and ADAR1 protect cells from PKR-mediated cell death
Finally, we generated cells with double KO (dKO) for PACT and PKR (Supplementary Fig. 8A). Knockdown of ADAR1 in dKO PACT/PKR cells failed to induce translation shutdown and stress granule formation (Fig. 8A, B and Supplementary Fig. 8B). More importantly, the depletion of PKR in PACT knockout cells completely prevented cell growth defect and cell death induced by the knockdown of ADAR1 (Fig. 8C–E). Taken together, these results demonstrate that either PACT or ADAR1 is required to prevent endogenous RNAs from triggering PKR-mediated translation arrest and cell death. Therefore, we propose that PACT and ADAR1 work together to sustain a threshold of tolerable dsRNA levels, permitting the presence of self-dsRNA in cells without triggering PKR-mediated translation shutdown and cell death (Fig. 9). During viral infection, dsRNA levels exceeding this protective threshold of PACT and ADAR1 result in PKR activation and subsequent translation arrest. In uninfected cells, PACT or ADAR1 alone can prevent PKR activation by self-dsRNA. However, the simultaneous deletion of both results in aberrant PKR activation, causing cell death (Fig. 9).

## Discussion
PKR activation in response to viral infection is essential for preventing viral replication and spread by inducing translation shutdown in host cells[8,12]. Given the harmful consequences of unintended PKR activation, it is crucial for cells to tightly regulate PKR activity, ensuring it is only triggered in response to viral infection[8,12]. Yet, many endogenous RNAs in cells can also act as substrates, mimicking viral RNAs and activating PKR in uninfected cells[8,19]. This underscores the importance of cells inhibiting self-dsRNA inducing PKR in the absence of viral infection. Indeed, aberrant or uncontrolled activation of PKR participates in the pathogenesis of several immune diseases and neurological disorders[63,64]. In this study, we found that cells employ two layers of protection to prevent the aberrant activation of PKR by endogenous dsRNAs. Using the CRISPR-Translate method[27], we identified that PACT and ADAR1 are essential factors restricting PKR activation both in response to viral infection and from self-dsRNA. Notably, the deletion of both PACT and ADAR1 results in synthetic lethality even in the absence of viral infection, highlighting their critical role in suppressing PKR activation by endogenous dsRNAs. Based on these findings, we propose that PACT and ADAR1 act as critical barriers to PKR activation by setting up a threshold of tolerable levels of endogenous dsRNA in cells without activating PKR. When dsRNA levels rise due to viral infection, PACT and ADAR1 become insufficient to prevent PKR binding to dsRNAs, which then triggers its activation and translation shutdown in virus-infected cells.

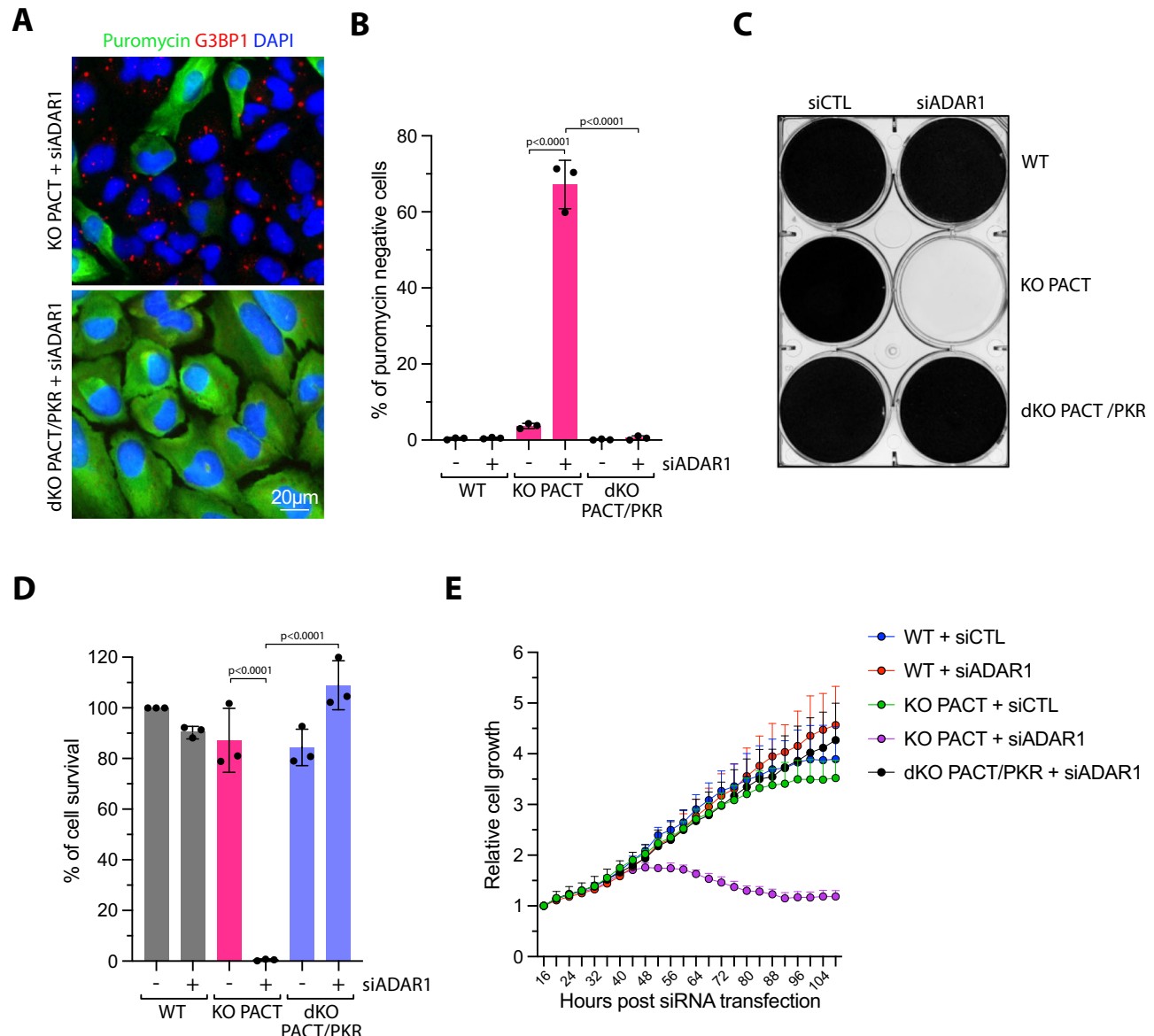

**Fig. 8 | The loss of PKR rescues cell viability after depletion of PACT and ADAR1. A** Representative immunofluorescence for puromycin and G3BP1 in U2OS PACT KO or PACT/PKR dKO cells knocked down with siRNA against ADAR1. **B** U2OS WT, PACT KO, or PACT/PKR dKO cells knocked down with siRNA control (siCTL) or against ADAR1 (siADAR1), and the levels of puromycin-negative cells (%) were quantified by immunofluorescence. Mean values ± SD (Number of biological replicates, $n$ = 3). $P$-values were calculated by two-way ANOVA. **C** Crystal violet staining showing the viability of U2OS WT, PACT KO, or PACT/PKR dKO cells transfected with siCTL or siADAR1. Cells were stained with Crystal violet 6 days following transfection with siRNA. **D** Indicated U2OS cell lines were knocked down with siCTL or siADAR1. Cell survival was quantified with Alamar blue cell viability assay 6 days following siRNA transfection. Mean values ± SD (Number of biological replicates, $n$ = 3). $P$-values were calculated by two-way ANOVA. **E** Indicated U2OS cell lines were transfected with siRNA control (siCTL) or against ADAR1 (siADAR1). Cell growth was then monitored starting at 16 h following siRNA transfection. 9 image fields were analyzed per well. Mean values ± SD (Number of biological replicates, $n$ = 3). Source data are provided as a Source Data file.

Herein, we employed an unbiased approach known as CRISPR-Translate, developed in our laboratory[27], to identify regulators of PKR. CRISPR-Translate is a FACS-based, genome-wide CRISPR-Cas9 knockout screening technique that uses translation levels as a readout[27]. By leveraging PKR-mediated translation shutdown, we applied CRISPR-Translate and discovered that, in addition to ADAR1, PACT is a crucial inhibitor of PKR in response to RNA viral infections. Consistent with this finding, a recent preprint manuscript from Dr. Sun Hur's team also reported a similar role of PACT inhibiting PKR activation mediated by endogenous dsRNAs in cells[49]. Initially identified as an activator of PKR[34,36], PACT's role in PKR regulation has since been controversial and remains debated[8]. Indeed, several studies have suggested that PACT suppresses PKR in contexts such as HIV infection[65–67] and in the

absence of the splicing factor TIA1[68]. Moreover, depletion of RAX, the mouse homolog of PACT, was shown to have reproductive and developmental defects caused in a PKR-dependent manner[69]. Moreover, the Hur laboratory manuscript suggested that PACT-dependent activation of PKR in vitro could have been caused by incomplete removal of RNA during the purification process of PACT. The authors found that extensive RNase treatment is required to properly remove RNAs that may affect the results[49]. Finally, mutations in PACT have been associated with early-onset dystonia (DYT16), which is caused by hyperactivation of PKR[64,70,71], further supporting the notion that PACT may function as a suppressor of PKR. Thus, thoroughly characterizing PACT's role in specific cellular contexts is crucial for better understanding its impact on PKR function.

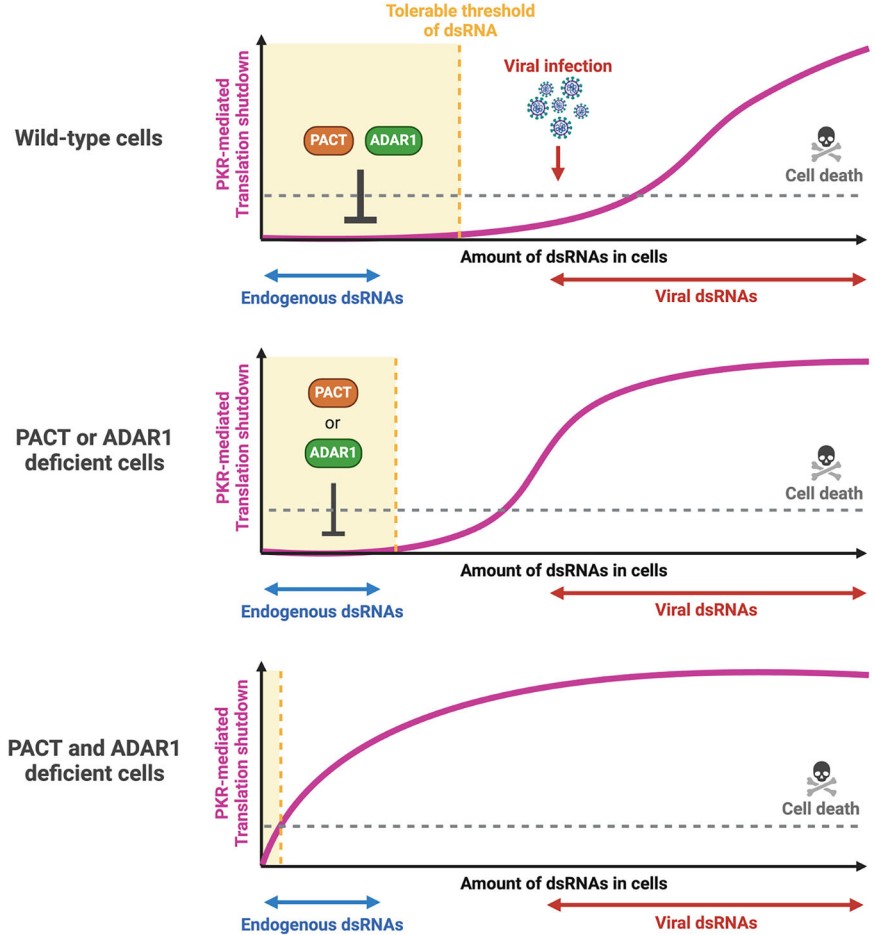

**Fig. 9 | Models illustrating how PACT and ADAR1 establish a threshold for tolerable dsRNA levels to prevent PKR activation.** PACT and ADAR1 collaborate to maintain a threshold of tolerable dsRNA levels, allowing cells to manage self-dsRNA without triggering PKR-mediated translation shutdown and cell death. During viral infection, however, the increased levels of dsRNA exceed the protective threshold set by PACT and ADAR1, leading to PKR activation. Either PACT or ADAR1 alone is sufficient to sustain this threshold, preventing self-dsRNA from activating PKR. Conversely, simultaneous loss of both PACT and ADAR1 results in uncontrolled PKR activation, causing rapid cell death. Created in BioRender. Buisson, R. (2025) https://BioRender.com/o10z759.

Mechanistically, our data suggested that PACT RNA binding is required to suppress PKR. PACT and PKR could compete for dsRNA binding in cells, or PACT may act as a barrier on dsRNA, preventing PKR scanning along dsRNA and decreasing the likelihood of dimerizing[49]. AlphaFold 3 modeling of PACT in complex with dsRNA revealed the formation of a PACT dimer that surrounds the dsRNA, potentially shielding the RNA from PKR binding. Consistently, PACT overexpression fully abrogated PKR activity across various cellular contexts, suggesting that elevated PACT levels shift the equilibrium toward PACT binding to dsRNA, even in the presence of high dsRNA levels, thereby preventing PKR activation. Both PKR and PACT bind similar inverted-repeat Alu elements in cells[19,49,72], further supporting the potential competition for these targets. Efficient PKR activation relies on scanning along dsRNA to facilitate dimerization, increasing the probability of encountering a second monomer and undergoing autophosphorylation[49]. Therefore, PACT does not need to saturate all available dsRNA but only needs to associate with a subset of binding sites to significantly reduce the chance of two PKR monomers interacting on a single dsRNA strand.

ADAR1 is another key enzyme in the human innate immune defense system, essential for preventing host-derived cytosolic dsRNAs from activating PKR in the absence of viral infection[19,22]. Mutations in ADAR1 are linked to Aicardi-Goutières syndrome (AGS)[73,74], a rare autoimmune disorder, and neurological diseases, further highlighting the importance of cells being protected against self-dsRNA-induced innate immunity. CRISPR-Translate also identified ADAR1 as a regulator of PKR following SeV infection, supporting previous studies that characterize ADAR1 as an inhibitor of PKR during RNA viral infections and from endogenous RNAs[19,22,29]. Knockout or knockdown of PACT or ADAR1 in cells is viable, indicating that depletion of either PACT or ADAR1 alone has minimal impact on PKR activation in unstressed cells. However, the depletion of both PACT and ADAR1 results in high PKR phosphorylation levels and rapid cell death, suggesting that PACT and ADAR1 have redundant roles in protecting cells from dsRNA-induced aberrant PKR activation. Importantly, the synthetic lethality phenotype can be completely rescued by knocking out PKR, indicating that the cell death resulting from the absence of PACT and ADAR1 is solely mediated by the aberrant activation of PKR. Based on these findings, we propose that PACT and ADAR1 share overlapping roles in inhibiting PKR, with each potentially serving as a backup for the other. This redundancy may help buffer stochastic variation in the expression of PACT and ADAR1, which can fluctuate depending on the cellular context and the stresses encountered by the cells. Furthermore, our findings indicate that the role of ADAR1 in limiting PKR activity is independent of its deaminase activity. This aligns with previous studies showing that ADAR1 prevents self-RNA-induced PKR activation primarily through its RNA-binding function rather than its deaminase capability[19,22,23]. However, we cannot fully rule out the possibility that both functions may play a role in preventing PKR activation. Utilizing an overexpression system to

complement ADAR1-deficient cells with a catalytically inactive mutant may help bypass the need for deaminase activity. Nevertheless, at endogenous levels, both functions could be essential, especially considering that ADAR1-p150, the cytoplasmic isoform of ADAR1, is expressed at lower levels compared to ADAR1-p110, the nuclear isoform. Further studies are needed to precisely pinpoint the specific domains of ADAR1 required to limit PKR activation in response to endogenous or viral dsRNAs.

Although PKR activation is critical to prevent viral replication in infected cells, many viruses have developed resistance mechanisms to inhibit PKR[14]. Influenza A virus (IAV) and vaccinia virus (VACV) encode viral dsRNA-binding protein NS1 and E3L proteins, respectively, to compete with PKR for dsRNA binding[75,76]. Toscana virus (TOSV) and Rift Valley fever virus (RVFV) promote PKR degradation through the expression of NS proteins[77–79]. The PKR pathway can also be hijacked by certain viruses, such as the Dengue and Zika viruses, to induce translation arrest and exploit host translational machinery to favor their own translation through a non-canonical mechanism[80,81]. These observations suggest that PKR regulation by PACT and ADAR1 may not necessarily result in a decrease in viral replication, but could also be beneficial for the virus, depending on the type and context. Although the primary goal of this study was not to investigate the effects of PACT on different types of virus replication, it is tempting to speculate that PACT could function as either a pro-viral or anti-viral protein through its regulation of PKR, depending on the cellular context and the type of viral infection. Indeed, PACT has been identified as a proviral factor for Dengue replication[82], similar to PKR[80], and as an anti-viral factor for IAV, HIV, and SARS-CoV2[67,83,84]. Viruses may exploit elevated PACT and ADAR1 levels or modulate their interaction with dsRNA to suppress PKR activation and prevent host cell death. Therefore, a deeper understanding of how host cells regulate PKR through PACT in response to viral infections is crucial for identifying viral resistance mechanisms.

## Methods

### Plasmids
PACT cDNA was obtained from Addgene (pENTR4_PRKRA; plasmid # 106110). pLenti-PACT-HA was generated by inserting PACT cDNA into a modified pLenti vector containing an HA-tag in the N-terminal using the Gateway Cloning System (Thermo Fisher Scientific). pLenti-PACT$^{EA}$ (K84E, K85E, K88A, K177E, K178A, K181A) was generated by site-directed mutagenesis. pInducer20-PACT-HA was generated by inserting the PACT-HA cDNA into the pInducer20 vector using the Gateway cloning system. ADAR1 cDNA was obtained from Addgene (pmGFP-ADAR1-p150; plasmid #117927) and subcloned in the pLenti vector. ADAR1-p150-E912A-eGFP was generated by site-directed mutagenesis.

### Cell culture
U2OS and HEK-293FT cells were cultured in DMEM supplemented with 10% FBS, 1% L-Glutamine, and 1% penicillin/streptomycin. A549 cells were maintained in DMEM / F12 GlutaMAX™-I supplemented with 10% FBS and 1% penicillin/streptomycin. Cell lines were purchased from either ATCC or Sigma-Aldrich. U2OS-derived cell lines were generated by infecting U2OS-derived cell lines were generated by infecting U2OS WT or PACT KO cells with lentivirus derived from pLenti or pInducer20 vectors expressing PACT-HA, PACT$^{EA}$-HA, or ADAR1-p150-eGFP and selected with G418 (850 μg/mL) for 6 days. RD cells were cultured in DMEM supplemented with 10% FCS.

### Viruses
Sendai virus (SeV) Cantell strain was propagated in 10-day-old embryonated chicken eggs for 48 h as previously published[85]. Allantoic fluid from the eggs was harvested, and the debris was removed by centrifugation for 25 min at 2600 × $g$ (4 °C). The supernatant was subjected to another round of centrifugation for 90 min at 27,000 × $g$

(4 °C), and the resultant pellet was resuspended in PBS supplemented with Ca$^{2+}$ and Mg$^{2+}$ (Thermo Fisher Scientific, #1404-133) and 1 mM EDTA. The pellet was sonicated and SeV viral titer was determined by plaque assay using Vero cells. Cells were plated and infected with SeV in 200 μL of serum-free medium for 1 h for adsorption at 37 °C. Then, Vero cells were overlaid with agarose (0.45% in culture media supplemented with 5 μg/mL acetylated trypsin). After 5 days, cells were fixed with trichloroacetic acid (10%) for 30 min, stained with crystal violet (0.1% crystal violet / 25% EtOH) for 5 min, and counted to determine viral titer. Sindbis virus (SINV) Ar-339 strain was purchased from ATCC (#VR-1585). SINV viral titer was determined by plaque assay using Vero cells. Cells were plated into 6-well dishes, and the adsorption of the virus was performed for 1 h at 37 °C in 200 μL of virus diluted in serum-free DMEM. Then, Vero cells were overlaid with agarose (0.45% in culture media). Poliovirus type 1 (PV) (Mahoney strain) was originally obtained from Dr. Eckard Wimmer at Stony Brook University, New York. PV viral titer was determined by plaque assay using HeLa cells. Cells were plated into 6 well dishes, and adsorption of the virus was performed for 30 min at room temperature in 200 μL of virus diluted in serum-free DMEM. Then, cells were overlaid with agarose (0.45% in culture media). EV-A71 was obtained from Dr. Shin-Ru Shih at Chang Gung University, Taiwan. A plasmid harboring the viral cDNA was used as a template for in vitro transcription[86]. Synthetic viral RNA was used to transfect RD cells. Individual infectious virus plaques were isolated and expanded to form a working stock. EV-A71 viral titer was determined by plaque assay using RD cells. Cells were plated into 6-well dishes, and adsorption of the virus was performed for 1H at 37 °C in 200 μL of virus diluted in serum-free DMEM. Then, cells were overlaid with agarose (0.45% in culture media).

### Viral infection
U2OS or A549 cells were infected with SeV, SINV, PV, or EV-A71 in serum-free medium at indicated M.O.I (Multiplicity of infection) at 37 °C for 1 h (SeV, SINV and EV-A71) or 22–25 °C for 30 min (PV) for adsorption. A culture medium supplemented with 10% FBS and 1% penicillin/streptomycin was added post-adsorption.

### Cell treatment
Poly(I:C) (InvivoGen, #tlrl-pic) and Poly(I:C)-Fluorescein (InvivoGen, #tlrl-picf) were transfected by forward transfection with Lipofectamine 2000 (Thermo Fisher Scientific, #11668019) at the indicated concentration and time according to the manufacturer's instructions. For Surface sensing of translation (SUnSET) Assay[37], U2OS, or A549 cells were treated with puromycin (10 μg/mL, MP Biomedicals #ICN10055210) for 15 min before fixation for analysis by immunofluorescence with an antibody against puromycin. U2OS PACT KO cells complemented with PACT-HA using the pInducer20 vector were treated with doxycycline (500 ng/mL) for 16- to 24 h prior to other treatments to induce protein expression.

### Lentivirus
All experiments involving the overexpression of PACT-HA, PACT$^{EA}$-HA, and ADAR1-p150-GFP were conducted using stable cell lines generated through lentiviral transduction of the corresponding constructs. Lentiviruses were generated using pLenti-PACT-HA, pLenti-PACT$^{EA}$-HA, pLenti-eGFP-ADAR1-p150, or pInducer20-PACT-HA vectors in combination with third-generation lentivirus packaging vectors: pMD2.G (Addgene #12259), pRSV-Rev (Addgene #12253), pMDLg/pRRE (Addgene #12251)[87]. The vectors were transfected in HEK-293T cells using Lipofectamine 2000 (Thermo Fisher Scientific, #11668027) according to manufacturer instructions. Two days following transfection, cell supernatant containing lentivirus was filtered, mixed 1:2 with target cell media and supplemented with polybrene (10 μg/ml). The next day, cells expressing PACT-HA or PACT$^{EA}$-HA were selected using G418 (850 μg/mL) for 6 days to obtain > 95% of the cell population

expressing PACT^WT or PACT^EA. Cells expressing PACT following infection with lentiviruses derived from the pInducer20-PACT-HA vector were subcloned by limited dilution to obtain > 95% of the cell population expressing PACT after doxycycline (500 ng/mL) treatment for 16–24 h. Cells expressing ADAR1-p150-eGFP were selected by FACS sorting.

## CRISPR-Translate

The CRISPR Knockout screening using the Brunello library[88] was performed following protocols provided by Addgene (Catalog #73179) and Dr. Feng Zhang lab[89] with adaptations[27]. For Brunello library lentiviral pool production, 293FT cells (ThemoFisher Scientific, # R70007) were seeded in 15 cm² culture dishes to have 70% confluency on the day of the transfection. The day after cell seeding, 15 μg of Brunello library plasmid (Addgene, #73179) and lentivirus packaging vectors (8 μg of pRSV-Rev, 8 μg of pMDL/pRRE, and 3 μg of pMD2.6) were transfected using calcium phosphate transfection method[90]. 48 h following transfection, cell supernatants were collected, filtered with 0.45 μm syringe filters (Genesse Scientific, #25-246), and frozen in 1.5 mL aliquots at − 80 °C.

The genome-wide CRISPR libraries were generated by transducing 140 million U2OS WT cells with a Brunello lentiviral pool at MOI of 0.3 to maintain at least 500x library coverage after puromycin selection. The cells were split once after reaching confluency during puromycin selection. Following 7 days of puromycin selection (0.75 μg/ml), 200 million cells were plated in 15 cm tissue culture dishes. The next day, the Brunello library harboring cells were infected with SeV at MOI = 3 for 14 h. The cells were shifted to methionine-free media for 30 min, followed by treatment with L-azidohomoalanine (AHA) (Vector Laboratory, #CCT-1066) at 25 μM for 30 min. The cells were collected and fixed with ice-cold 70% ethanol overnight. Next, the cells were labeled with a 488-tagged alkyne probe (Vector Laboratories, #CCT-1277-1) using click-It reaction (Vector Laboratories, #CCT-1263) according to the manufacturer's protocol. The cells were then sorted on a FACS Aria Fusion into two populations: 488 positive cells and 5% of the bottom 488 negative cells. 488 negative cells were subjected to a second round of sorting to eliminate any false 488-negative cells. Genomic DNA was extracted from both sorted cell populations, and sgRNAs were amplified using P5 primers with different numbers of stagger regions pooled together (for sequencing diversity) and P7 primers (Supplementary Data 2) with unique barcode sequences using Q5 High-Fidelity DNA Polymerase (New England Biolabs, #M0491) under the following PCR condition: an initial denaturation at 98 °C for 30 s, followed by 10 s at 98 °C, 30 s at 65 °C, 30 s at 72 °C for 32 cycles, and a final extension at 72 °C for 10 min. PCR products were gel extracted (Qiagen, #28706) and sequenced on a Novaseq 6000 platform (UCI Genomics High-Throughput Facility (GHTF)). MaGeck analysis was performed to find enriched gRNAs in the 488-negative population relative to the 488-positive population (Supplementary Data 1)[31].

## Synthetic lethality CRISPR knockout screen

The survival screen consisted of transducing 140 million U2OS WT or PACT KO cells with a Brunello lentiviral pool at MOI of 0.3 to maintain at least 500x library coverage after puromycin selection. The cells were split once after reaching confluency during puromycin selection. Following 3 days of puromycin selection (1 μg/ml), to maintain a 500-fold library coverage, 40 million cells of each cell line (U2OS WT or PACT KO) were plated in 15 cm tissue culture dishes and allowed to grow for 7 days. Genomic DNA was extracted using the Quick-DNA Midiprep Plus Kit (ZymoResearch, #D4075), and sgRNAs were amplified using P5 primers with different numbers of stagger regions pooled together (for sequencing diversity) and P7 primers (Supplementary Data 2) with unique barcode sequences using Q5 High-Fidelity DNA Polymerase (New England Biolabs, #M0491) under the following PCR condition: an

initial denaturation at 98 °C for 30 s, followed by 10 s at 98 °C, 30 seconds at 65 °C, 30 s at 72 °C for 32 cycles, and a final extension at 72 °C for 10 min. PCR products were gel extracted (Qiagen, #28706) and sequenced on a Novaseq 6000 platform (UCI Genomics High-Throughput Facility (GHTF)). MaGeck analysis was performed to identify gRNAs depleted in PACT KO cells (Supplementary Data 1). P-values were calculated using a negative binomial model with a likelihood ratio test, and gene-level significance was determined using Robust Rank Aggregation (RRA), which was implemented in MAGeCK. Multiple testing correction was applied using the Benjamini-Hochberg false discovery rate (FDR) method.

## RNA interference

siRNA transfections were performed by reverse transfection with Lipofectamine RNAiMax (Thermo Fisher Scientific, #13778150). siRNAs were purchased from Thermo Fisher Scientific (Silencer Select siRNA). Cells were transfected with poly(I:C) or infected with viruses 40 h after siRNA transfection (4–8 nM) if not indicated otherwise. To increase knockdown efficiency, two siRNA targeting PKR or ADAR1 were transfected together. The sequences of the siRNAs used in this study are listed in Supplementary Data 2.

## Antibodies

The antibodies used in this study are listed in Supplementary Data 2.

## CRISPR-Cas9 Knockout cells

PKR and ADAR1-p150 CRISPR-Cas9 knockout U2OS cell line was performed by transfection with Lipofectamine CRISPRMAX of TrueGuide Synthetic CRISPR gRNA (Thermo Fisher Scientific, #CMAX00003) and TrueCut Cas9 Protein v2 (ThermoFisher Scientific, #A36498) according to the manufacturer's instructions. CRISPR gene editing efficiency was verified using the GeneArt Genomic Cleavage Detection kit (A24372; Thermo Fisher Scientific). PACT KO cells and PKR KO-derivative knockout cell lines were generated by transfecting cells with the pSpCas9(BB)-2A-GFP (PX458) plasmid containing gRNAs targeting each gene with FuGENE 6 Transfection Reagent (E2691; Promega). 24 h after transfection, GFP + cells were sorted and selected. For every target, three or more independent clones were generated. gRNA sequences used in this study are listed in Supplementary Data 2.

## Immunofluorescence

Cells were fixed with paraformaldehyde (3% paraformaldehyde and 2% sucrose in 1xPBS) for 20 min, washed twice with 1 × PBS, and cells were permeabilized with a permeabilization buffer (1 × PBS and 0.2% Triton X-100) for 5 min. Subsequently, cells were washed twice with 1xPBS, and blocked in PBS-T (1xPBS and 0.05% Tween-20) containing 2% BSA and 10% milk for 1 h. Cells were then incubated with the primary antibody diluted in 1xPBS-T containing 2% BSA and 10% milk at room temperature for 2 h. Coverslips were washed three times with PBS-T before incubation (1 h) with the appropriate secondary antibodies conjugated to fluorophores (Alexa-488 or Cy3). After three washes with PBS-T, cells were stained with DAPI (0.5 μg/mL, MilliporeSigma #D9542), and the coverslips were mounted using slow-fade mounting media (Thermo Fisher Scientific, # S36936). Images were captured using a Leica DMi8 THUNDER microscope.

## RNA fluorescence in situ hybridization (RNA FISH)

Cells were fixed with paraformaldehyde (3% paraformaldehyde and 2% sucrose in 1 × PBS nuclease-free) for 10 min, washed twice with 1 × PBS, and permeabilized with a permeabilization buffer (1 × PBS and 0.2% Triton X-100) for 5 min. Subsequently, cells were washed twice with nuclease-free 1 × PBS and then incubated with the primary antibody diluted in nuclease-free 1 × PBS containing 2% BSA at room temperature for 2 h. Coverslips were washed three times with PBS-T before incubation (1 h) with the appropriate secondary antibodies conjugated

to fluorophores. The stained cells were washed three times with PBS-T and then fixed again with paraformaldehyde (3% paraformaldehyde and 2% sucrose in nuclease-free 1 × PBS) for 10 min, followed by 1 h incubation in 70% ethanol at room temperature. Next, cells were washed in FISH wash buffer (10% formamide and 2X SSC (Saline Sodium Citrate)) for 5 min. The FISH probes (12.5 μM stock) diluted 1:100 in FISH hybridization buffer (10% formamide, 2X SSC, and 10% (wt/vol) dextran sulfate) were added to each coverslip placed in a petri dish containing a wet paper towel and then covered with parafilm to create a humidity chamber. Hybridization was performed at 37 °C for 16 h. The cells were then washed once in FISH wash buffer for 30 min. Finally, cells were stained with DAPI (5 μg/mL, MilliporeSigma #D9542) in 2X SSC for 5 min before the coverslips were mounted using slow-fade mounting media (Thermo Fisher Scientific, #S36936). Images were captured using a Leica DMi8 THUNDER microscope. The FISH probes used in this study were designed to target positions 14,944 to 15340 of the SeV genome (GenBank: AB855654.1) as previously described[38] and were labeled with Quasar 670 (Biosearch Technologies).

### Crystal violet staining assay
U2OS or A549 cells (60,000 cells per well) were plated in 6 well plates and reverse-transfected with the indicated siRNA. Six days following siRNA transfection, cells were fixed with paraformaldehyde (3% paraformaldehyde and 2% sucrose in 1 × PBS) for 20 min, washed twice with 1 × PBS, cells were stained with a solution of 0.5% crystal violet solution (Fisher Chemicals, #C581-25) and 30% methanol for 2 h at room temperature, and washed thoroughly with 1 × PBS to remove all the crystal violet solution.

### Alamar blue cell viability assay
Cells were plated in a 96-well plate at a density of 5000 cells/well to obtain 90 to 95% confluency in control cells after 6 days. Cells were reverse-transfected with the indicated siRNAs. Cell viability was assessed 6 days post-transfection using Alamar Blue Cell Viability Reagent (Thermo Fisher Scientific, #DAL1025). 20 μl of Alamar Blue was added to each well and incubated for 4 h. The fluorescence levels (excitation 565 nm/emission 590 nm) were then measured using a Varioskan LUX Multimode Microplate Reader microplate reader (Thermo Fisher Scientific, #VLBL00GD2).

### Quantitative reverse transcription PCR (RT-qPCR)
Total RNA was extracted from cells using Quick-RNA MiniPrep Kit (Zymo Research, #R1055) according to the manufacturer's instructions. Following extraction, total RNA was reverse transcribed using the High Capacity cDNA Reverse Transcription Kit (Thermo Fisher Scientific, #4368813). RT products were analyzed by real-time qPCR using SYBR Green (PowerUp SYBR Green Master Mix, Thermo Fisher Scientific, #A25743 in a QuantStudio 3 Real-Time PCR detection system (Thermo Fisher Scientific). For each sample tested, the levels of indicated mRNA were normalized to the levels of Actin mRNA. The primers used in this study are listed in Supplementary Data 2.

### AlphaFold 3 modeling
AlphaFold 3 modeling was performed using the AlphaFold webserver (https://alphafoldserver.com/)[48]. The dsRNA sequences used in the predictions were derived from sequences shown to stimulate PKR autophosphorylation[15] (GGAGAACUUCAUGCCCUUCGGAUAAGGACU and AGUCCUUAUCCGAAGGGCAUGAAGUUCUCC). Local folding accuracy was evaluated by pLDDT score plotted for each atom in the model, with the AlphaFold3 recommended score thresholds of 0, 50, 70, and 90 for Very Low, Low, High, and Very High confidence, respectively. Interdomain interaction predictions were evaluated based on the predicted aligned error (PAE) score[48] as visualized either by the PAE Viewer webserver[91] or using a Python script to identify the

average interface PAE score[92]. Congruence between the five models predicted by AlphaFold 3 for each query was performed using the Matchmaker function in ChimeraX[93].

### Proliferation assays
Cells were transfected with siRNA control (siCTL) or against siADAR1 (siADAR1) by reverse transfection with Lipofectamine RNAiMax (Thermo Fisher Scientific, #13778150) and were seeded at a low dilution in triplicate in a 6-well dish (60,000 cells per well). Cell media containing transfection reagent was then replaced with fresh cell media 16 h after siRNA transfection. Cell growth was monitored over time using a BioTek Cytation 5 Cell Imaging Multimode Reader (Agilent) equipped with BioTek BioSpa 8 Automated Incubator (Agilent). Nine images were taken every 4 hours in each well, and the number of cells by images were determined using BioTek Gen5 Software (Agilent).

### Statistics and reproducibility
Electrophoresis gels (Figs. 2a–f, 4d, g, h, 6a, c, 7a, b, f; Supplementary Figs. 1a–c, 2a, b, 5a, 6a–c, 6f–I, 7a, 8a) and immunofluorescence panels (Figs. 1c, 3a, c, 6b, d, 8a; Supplementary Figs. 1e, f, 2c, 7b, c) were repeated at least three times, and representative images are shown in this paper.

### Reporting summary
Further information on research design is available in the Nature Portfolio Reporting Summary linked to this article.

## Data availability
CRISPR knockout screen data generated in this study have been deposited in NCBI's Sequence Read Archive (SRA) with the Bioproject Accession number PRJNA1222466. Source data are provided in this paper.

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

## Acknowledgements

We thank Dr. Christopher Halbrook for providing access to the BioTek Cytation 5 Cell Imaging Multimode Reader. We thank Casey Johnson, Fiona Law, and Dr. Melanie Oakes for their technical assistance, and Dr. Ivan Marazzi for his helpful discussions. pENTR4_PRKRA was a gift from Thomas Tuschl (Addgene #106110). pmGFP-ADAR1-p150 was a gift from Kumiko Ui-Tei (Addgene plasmid # 117927). The human Brunello CRISPR knockout pooled library was a gift from David Root and John Doench (Addgene #73179). L.M. was supported by a Center for Virus Research Graduate Fellowship funded by the UCI Division of Graduate Studies. Salary support for P.O. was provided by a California Institute for Regenerative Medicine (CIRM) stem cell biology training grant (TG2-01152) and an EMBO Postdoctoral fellowship (ALTF 213-2023). G.S. and A.S. were supported by the National Institutes of Health Research Supplements to Promote Diversity in Health-Related Research (R37-CA252081-S1; -S2). S.O. is a Dr. Lorna Calin Scholar and was supported by the Faculty Mentor Program and a Graduate Dean's Dissertation Fellowship from the University of California, Irvine. R.B. was supported by the National Institutes of Health (R37-CA252081, R21-AI185033, and

R21-ESO36190) and a Research Scholar Grant (RSG-24-1249960-01-DMC) from the American Cancer Society. B.L.S. was supported by the National Institutes of Health (R01-AI155962). D.S. was supported by the Natural Sciences and Engineering Research Council of Canada Discovery Grant (RGPIN–2024-05998) and a Next Generation Scientist Scholarship (#26002) from the Cancer Research Society.

## Author contributions

L.M., G.S., P.O., A.S., S.O., A.G., J.L., D.D., and E.B. performed all the experiments. A.B and B.L.S provided critical reagents, laboratory access to performed experiments with poliovirus, and input on the project. D.S. performed the AlphaFold 3 modeling. L.M. and R.B. conceived the study, designed the experiments, and wrote the paper. R.B. oversaw the project, and all the authors contributed to manuscript editing.

## Competing interests

The authors declare no competing interests.
