## [Transparent Peer Review file · Nature Communications]

Cooperative Role of PACT and ADAR1 in Preventing Aberrant PKR Activation by Self-Derived Double-Stranded RNA

Corresponding Author: Professor Rémi Buisson

Version 0:

Reviewer comments:

Reviewer #1

(Remarks to the Author)

This manuscript addresses the mechanisms by which PKR is regulated to prevent PKR activation by endogenous dsRNAs. The main conclusion is that PACT, in tandem with ADAR1, inhibits PKR activity in mammalian cells, both under baseline conditions and in the context of antiviral signaling, which is supported by several observations. First, PACT-mediated inhibition of PKR translation arrest is dependent on the functionality of PACT's dsRNA binding domains. Second, combined depletion of PACT and ADAR1 is lethal in uninfected cells, which correlates with enhanced PKR activation and eIF2 α phosphorylation under these conditions.

The role of ADAR1 in suppressing innate immune activation by endogenous dsRNAs has been described previously (PMID: 29395325, PMID: 39146181). However, this work addresses the issue of whether PACT plays a stimulatory or inhibitory role in PKR regulation, and is consistent with findings by other groups (PMID: 24020926). Overall, the work is well done and could be published in Nature Communications after the comments below are addressed.

This review is from Roy Parker and I would be willing to clarify these comments for the authors if needed.

1) The key experiment demonstrating that PACT-mediated suppression of PKR is dsRNA-dependent is shown in Figure 4G. However, the PACTEA mutant with impaired dsRNA binding is only expressed in the context of viral infection, maintaining the possibility that other host-virus mechanisms involving PACT, and not the binding of PACT to dsRNA alone, may be responsible for PKR inhibition. This possibility is supported by enhanced translation arrest in PACTEA-expressing cells compared to the PACT KO condition. Given this, it would strengthen the manuscript to perform the same experiment using dsRNA stimulation sans live infection, such as with poly(I:C) transfection.

2) In Figure 6, it remains possible that total PKR levels are changing (since only western blots for p-PKR are shown). Can the authors address if PKR levels are changing when PACT and/or ADAR1 are depleted?

3) The dataset of other genes affecting PKR activation should be included in a supplemental excel file as these will spawn experiments in multiple labs.

4) Although I would not require for publication, the authors have an easy opportunity to test the role of phosphorylation sites in PACT that have been suggested to be required for PACT-based activation of PKR. It would clarify this issue to make the phosphomimetics at these sites in their PACT rescue construct and determine if these still repress PKR activation or instead activate PKR.

5) Although I would not require for publication: Given that PACT is thought to bind PKR, and my quick AlphaFold 3 use identifies possible interactions between PACT and PKR, it would strengthen the manuscript to determine if PACT-PKR physical interactions play a role in the regulation or not.

Minor Comments:

6) It is notable that eIF2 phosphorylation is not examined in response to poly(I:C) treatment, which is presumably due to RNase L activation activating eIF2a phosphorylation (PMID: 31494035 & 38823018). To note this issue here would make the manuscript more scholarly for the informed reader.

7) Prior work on ADAR1 siRNAs have also shown that this can lead to a low level of spontaneous PKR activation (PMID: 34397095). This highlights how having two different mechanisms to limit PKR activation that work at different molecular levels can buffer stochastic variation in gene expression (which is why cells often require multiple independent regulatory systems), which might be worth noting in the discussion (although this is solely up to the authors).

Reviewer #2

(Remarks to the Author)

This manuscript leverages a CRISPR-translate screening developed by the authors to identify PACT and ADAR1 as regulators of PKR activation and consequent translation arrest induced by both viral and self-RNA. The study supports and complements a recent publication describing a role for PACT in regulating PKR response to endogenous RNAs. Using a combination of gene knockout, gene knockdown, protein complementation, WB, and RNA virus infection assays, the authors demonstrate that PACT and ADAR1 play necessary and redundant roles in preventing PKR activation. Additionally, the authors used AlphaFold to model the structural interaction between PACT and dsRNA, identifying the RNA binding sites of PACT and confirmed that PACT's binding activity is essential for regulating PKR activation. These findings provide new insights into how cells tolerate self-RNA but not viral RNA, shedding light on mechanisms that prevent aberrant immune activation. A few issues need to be resolved to fully support the conclusion.

Major issues:

1. To control for differences in the rate of infection, staining for viral proteins should be added to all experiments. Interpretation of several results is limited without a control for infection.
2. What is the relevance of quantifying number of foci per cell? As discussed by the authors and recently shown in the context of SeV infection (PMID: 37983241), SG are dynamic appearing/growing/disappearing at different times during infection. To better support the conclusions on figure 3 the authors should label infected cells in each condition and quantify SG+ cells based on the number of infected cells.
3. The main conclusion of the manuscript (Figures 6/7) is that cells can tolerate self-RNA due to PACT / ADAR1 interference with PKR activation. To support this conclusion, the authors need to directly demonstrate binding of PACT/ ADAR to endogenous RNAs and test if high levels of endogenous RNA overcome this blockage. Virus infection impacts several cellular processes. The proposed model is indeed interesting but must be directly validated.
4. A discussion of how a virus infection would impact the interaction of PACT /ADAR with host RNAs and the consequences for cell survival is needed.

Minor issues:

1. Figure 1 SeV infection time does not match in results and legend (14 h vs 24h).
2. Figure 1A: Switching the positions of 488+ and 488- to align with Figure 1B would help readers to understand the data easily.
3. Line 129-130: "transduction" should be "infection", and I'd like to know which cell lines are being compared.
4. Figure 1C: The puromycin neg cells percentage in "KO PACT, SeV infection" is about 57.7% (15/26), which is much higher than the corresponding data shown in Figure 1D. What is the explanation for this?
5. Figure 2: It seems clear that PACT inhibition of PKR activity is more effective in the context of poly I:C than SeV infection. A discussion on this observation should be included.
6. The PACT complementation was done by lentivirus transduction, which should be better described. It was difficult to figure out the method from the current description (some people complement by transfection).
7. Figure 4D: a qPCR data for SeV genome replication is necessary to show SeV infection of different MOI.
8. Figure 4D and E use overexpression of PACT with different levels of viral dsRNA. Since the question is whether different expression of PACT would influence PKR activation by viral RNA, I'd like to see same amount of dsRNA (same MOI of SeV infection) but different amount of PACT plasmids transfection in PACT KO cells to validate the conclusions. SeV Cantell contain defective interfering particles that will reduce viral replication and the amount of dsRNA available, therefore the experiment is not conclusive as presented.
9. Discuss whether acute and persistent infections could be related to this, considering the differences in dsRNA levels between them.

Line 31: should read identified
Line 32: deficient on...
Line 255: to test this hypothesis...
In general needs editing
What is synthetic lethality?

Reviewer #3

(Remarks to the Author)

This study investigates cellular mechanisms that regulate activity of the innate immune sensor protein kinase R (PKR). They address important fundamental questions about how PKR is regulated to distinguish self double-stranded dsRNA (dsRNA) from viral RNA. They describe new functions for the protein PACT, suggesting it might interfere with PKR activation in response to viral infection. They further demonstrate that PACT cooperates with ADAR1 to suppress PKR activation from self-dsRNAs in uninfected cells. They propose that PACT and ADAR1 together act as a barrier against PKR activation to generate a threshold for levels of endogenous dsRNA that can be tolerated in cells. In elegant and rigorous experiments they first identify PACT through a CRISPR screen for suppressors of translation arrest during RNA viral infection. They then show that PACT inhibits PKR phosphorylation during RNA viral infection by immunoblotting in knockout cells. They demonstrate that PACT can suppress PKR-mediated arrest of protein translation and formation of stress granules in response to virus infections. Structural modeling is used to investigate interaction with dsRNA, and they suggest that higher PACT expression can inhibit PKR activation by protecting the dsRNA. A second CRISPR screen in PACT knockout cells identifies ADAR1 as synthetically lethal, and they suggest they act synergistically to suppress PKR activation. There are just a few key experiments that could be done to complement findings and further support conclusions. Overall, the study makes interesting observations that offer new perspectives to understand the activation of PKR in response to self-RNAs and how these dsRNA-binding proteins participate in the modulation of the antiviral response regulated by PKR. Experiments are well executed and described. It is a significant contribution that will be appreciated by those working on innate immune sensors and dsRNA binding proteins.

Specific Comments:

- 1) In Figure 1 they employ an elegant technique called CRISPR-Translate to screen a library to identify factors that regulate PKR and host translation shutoff. This identified PACT. The authors could further explore and describe the other hits that past their threshold that fall between PACT and ADAR1. The translational inhibition observed in Figures 1C-1E) show that it is detected in only a small proportion (around 10%) of SeV-infected cells at 24hpi. Why is this the case and does this suggest other proteins are also involved?
- 2) They suggest that PACT inhibits PKR primarily through its RNA-binding activity. Although data linking PACT overexpression to inhibition of PKR phosphorylation are compelling, further evidence is necessary to support that PACT interferes with PKR binding to dsRNA. However, they do not experimentally demonstrate that overexpression of PACT prevents PKR binding to dsRNA. Additional experiments could be done to support comments in lines 274 and 275 by demonstrating competition in vitro or Co-IP experiments using dsRNA antibodies (J2 or 9D5) testing whether PKR pulldown is decreased in cells overexpression PACT. In addition, pl:C transfection was shown to induce cytoplasmic dsRNA-induced foci (dRIFs) where PKR, PACT and ADAR1 accumulate (PMCID: PMC9388085). The authors could explore how the overexpression of PACT (and ADAR1) affects the recruitment of PKR to these sites of dsRNA accumulation. In addition, if it is really through a threshold model, they should be able to generate inducible systems to complement the knockout and demonstrate a dose response to effects on PKR activation.
- 3) If ADAR1 and PACT have redundant roles (as argued in lines 325-326), it should be shown by rescue in the phosphorylation of PKR in PACT KO cells following overexpression of ADAR1. It is shown that ADAR1 or p150-ADAR1 are sufficient to prevent PKR activation by endogenous RNAs. This should be demonstrated in PACT KO cells during SeV infection or pl:C transfection. This should be done by overexpressing ADAR1, p150-ADAR1 or catalytic inactive mutant (ADAR1-p150-E912A).

Minor Issues

- 1) Statistics: In all figures where statistic tests were performed, two-tailed t-test were used. In this approach looking for decrease in translation as an example, when comparing two groups, one-tailed tests should be applied. Authors however also compare different conditions. For these, analysis of variance (ANOVA) should be applied rather than t-tests for selected conditions in the same dataset.
- 2) Line 129. They use the word "transduction" when instead they are referring to infection.
- 3) Line 128-132. Authors picked 14h post infection with SeV as the best time point to screen for AHA positive and negative cells. Authors could provide a time course showing PKR phosphorylation over time in U2OS cells infected with SeV as well as a control for showing infected cells using a viral protein as marker.
- 4) Line 193 (Figure 2D and 2E). Both control viral proteins are marked as VP1 in W B panels. Is this the correct designation?
- 5) Line 216. PKR structural modeling is not explored in this section but is mentioned in the title.
- 6) Line 334 (Supplementary Fig 6F). Short and long exposure panels are inverted for ADAR1.
- 7) Line 357-374. Discussion Paragraph 1 lacks references.
- 8) Line 370. Using the term "cellular sensors" imply PACT and ADAR1 are sensing dsRNAs which will lead to a downstream response. In the paper it is argued that these proteins rather bind to dsRNA to prevent PKR activation in response to self-RNAs or its overactivation during viral infections.

Version 1:

Reviewer comments:

Reviewer #1

(Remarks to the Author)

The authors have addressed all of my comments and I am supportive of publication.

Reviewer #2

(Remarks to the Author)

We thank the authors for a wonderful job in addressing the concerns.

Reviewer #3

(Remarks to the Author)

The authors have done a commendable job in addressing all our prior critiques and responding to all reviewer comments. This work makes a nice contribution to the field.

REVIEWER COMMENTS

Reviewer #1 (Remarks to the Author):

This manuscript addresses the mechanisms by which PKR is regulated to prevent PKR activation by endogenous dsRNAs. The main conclusion is that PACT, in tandem with ADAR1, inhibits PKR activity in mammalian cells, both under baseline conditions and in the context of antiviral signaling, which is supported by several observations. First, PACT-mediated inhibition of PKR translation arrest is dependent on the functionality of PACT's dsRNA binding domains. Second, combined depletion of PACT and ADAR1 is lethal in uninfected cells, which correlates with enhanced PKR activation and eIF2 α phosphorylation under these conditions.

The role of ADAR1 in suppressing innate immune activation by endogenous dsRNAs has been described previously (PMID: 29395325, PMID: 39146181). However, this work addresses the issue of whether PACT plays a stimulatory or inhibitory role in PKR regulation, and is consistent with findings by other groups (PMID: 24020926). Overall, the work is well done and could be published in Nature Communications after the comments below are addressed.

This review is from Roy Parker and I would be willing to clarify these comments for the authors if needed.

Dear Dr. Parker,

Thank you very much for your constructive suggestions and for supporting the findings of our manuscript! We have now addressed all your comments and suggestions.

1) The key experiment demonstrating that PACT-mediated suppression of PKR is dsRNA-dependent is shown in Figure 4G. However, the PACTEA mutant with impaired dsRNA binding is only expressed in the context of viral infection, maintaining the possibility that other host-virus mechanisms involving PACT, and not the binding of PACT to dsRNA alone, may be responsible for PKR inhibition. This possibility is supported by enhanced translation arrest in PACTEA-expressing cells compared to the PACT KO condition. Given this, it would strengthen the manuscript to perform the same experiment using dsRNA stimulation sans live infection, such as with poly(I:C) transfection.

This is a very important point! We have now transfected cells with poly(I:C) and monitored PKR phosphorylation levels in PACT KO cells complemented with either PACT-WT or the PACT-EA mutant. Expression of PACT-WT fully suppressed PKR activation following poly(I:C) transfection, whereas PACT-EA failed to suppress PKR phosphorylation levels (**New Figure 4H**), demonstrating that PACT's dsRNA binding is required for suppressing PKR activation. Similar results were observed in PACT KO cells infected with SeV and complemented with either PACT-WT or PACT-EA (**New Figure 4G and supplementary Figure 6H**), further confirming that PACT-mediated suppression of PKR is dsRNA-dependent, regardless of whether the stimulus is an RNA virus or dsRNA alone. Additionally, we have repeated the experiment initially shown in **Figure 4I**, now using stable cell lines expressing either PACT-WT or PACT-EA (This experiment depicted in this panel was previously performed after the transient transfection of PACT-WT or PACT-EA and was the only experiment conducted with transient transfection rather than stable cell lines). Using our new stable cell lines, we now show that PACT-EA expression did not impact the translation arrest levels compared to PACT KO cells upon SeV infection (**New Figure 4I**). Finally, we performed additional experiments to show

that PACT WT, but not PACT-EA, suppresses PKR activation and restores cell survival in PACT knockout and ADAR1 knockdown cells (**New Figures 5F-G, 6C, and 6E**). Thank you for suggesting these experiments

2) In Figure 6, it remains possible that total PKR levels are changing (since only western blots for p-PKR are shown). Can the authors address if PKR levels are changing when PACT and/or ADAR1 are depleted?

We have now reblotted the cell extracts used to generate **Figure 6A** with a PKR antibody. We observed a slight decrease in the total PKR level in PACT-KO cells that underwent ADAR1 knockdown. This decrease is likely due to the hyperphosphorylation of PKR, which leads to a smear in the blot, or the phosphorylation event that masks the antibody's recognition of PKR. This result contrasts with the slight increase in total PKR observed after SeV infection (**Figure 2B**), which is mediated by the IFN response triggered by SeV infection (PKR is an interferon-stimulated gene (PMID: 30686999)).

3) The dataset of other genes affecting PKR activation should be included in a supplemental excel file as these will spawn experiments in multiple labs.

We had included a list of all identified genes and the rankings from our two CRISPR-Cas9 screens in an Excel document but forgot to add a reference to this document in the manuscript. We apologize for not including a reference to this document in our manuscript. We have now added it to the results section (**Supplementary Data 1**).

4) Although I would not require for publication, the authors have an easy opportunity to test the role of phosphorylation sites in PACT that have been suggested to be required for PACT-based activation of PKR. It would clarify this issue to make the phosphomimetics at these sites in their PACT rescue construct and determine if these still repress PKR activation or instead activate PKR.

During the development of this project, we tested the effects of PACT phosphorylation mutations (S18A, S167A) and mimics (S18D, S167D, and S18D/S167D) that are located close to or within the RNA binding domains of PACT. We found that all of these mutations restored translation in PACT-KO cells knocked down for ADAR1, similar to wild-type PACT (**New Supplementary Figure 6E**). This demonstrates that these PACT phosphorylation mutations or mimics can still repress PKR activation. However, we cannot rule out the possibility that phosphorylation mimics do not fully replicate the native phosphorylation state of PACT or that the overexpression of PACT bypasses this regulation by phosphorylation events. In addition, other phosphorylation sites that were previously identified in the dimerization domain would still need to be tested.

5) Although I would not require for publication: Given that PACT is thought to bind PKR, and my quick AlphaFold 3 use identifies possible interactions between PACT and PKR, it would strengthen the manuscript to determine if PACT-PKR physical interactions play a role in the regulation or not.

Our AlphaFold3 modeling did not predict with high confidence any direct interaction between PACT and PKR (**Rebuttal Figure 1A**). The interaction between PACT and PKR was only

observed indirectly through dsRNA binding. Consistent with this result, we found that the interaction between PACT and PKR following PACT immunoprecipitation was predominantly mediated through RNA. Treatment of cell extracts with RNase A suppressed most of the interaction between the two proteins (**Rebuttal Figure 1B**). Furthermore, a recent accompanying preprint from Dr. Sun Hur's lab (Ahmad et al. BioRxiv: <https://doi.org/10.1101/2024.10.23.619951>) reported only a weak protein-protein interaction between purified PACT and PKR. This interaction was mediated by the $\beta 3$ strand of DRBD1 and DRBD2 domains of PACT, and mutations in these domains were shown to impact PACT's ability to regulate PKR. Since Dr. Sun Hur's lab has already investigated the role of the interaction between PACT and PKR, we did not focus our study on this aspect.

Rebuttal Figure 1: A. Predicted Aligned Error (PAE) matrix of dimeric PACT (residues 1–313) with dimeric PKR. The black lines indicate the molecule's boundaries. **B.** Immunoprecipitation of PACT in the presence or absence of RNaseA. The interaction between PACT and PKR was analyzed by WB.

Minor Comments:

6) It is notable that eIF2 α phosphorylation is not examined in response to poly(I:C) treatment, which is presumably due to RNase L activation activating eIF2 α phosphorylation (PMID: 31494035 & 38823018). To note this issue here would make the manuscript more scholarly for the informed reader.

The difference in eIF2 α phosphorylation was very weak between WT and PACT KO cells transfected with poly(I:C), likely as suggested by the reviewer due to the activation of RNase L in cells transfected with poly(I:C). For the same reason, we did not monitor translation levels with puromycin labeling after poly(I:C) transfection due to the role of RNase L inhibiting translation. We have now added a sentence in the result section to comment on this specific point.

7) Prior work on ADAR1 siRNAs have also shown that this can lead to a low level of spontaneous PKR activation (PMID: 34397095). This highlights how having two different mechanisms to limit PKR activation that work at different molecular levels can buffer stochastic variation in gene expression (which is why cells often require multiple independent regulatory systems), which might be worth noting in the discussion (although this is solely up to the authors).

Thanks! We have now added a sentence to reflect this concept in the discussion of the manuscript.

Reviewer #2 (Remarks to the Author):

This manuscript leverages a CRISPR-translate screening developed by the authors to identify PACT and ADAR1 as regulators of PKR activation and consequent translation arrest induced by both viral and self-RNA. The study supports and complements a recent publication describing a role for PACT in regulating PKR response to endogenous RNAs. Using a combination of gene knockout, gene knockdown, protein complementation, WB, and RNA virus infection assays, the authors demonstrate that PACT and ADAR1 play necessary and redundant roles in preventing PKR activation. Additionally, the authors used AlphaFold to model the structural interaction between PACT and dsRNA, identifying the RNA binding sites of PACT and confirmed that PACT's binding activity is essential for regulating PKR activation. These findings provide new insights into how cells tolerate self-RNA but not viral RNA, shedding light on mechanisms that prevent aberrant immune activation. A few issues need to be resolved to fully support the conclusion.

We thank the reviewer for appreciating the novelty of our work and suggesting important experiments to further support our model. We have now addressed all the comments and suggestions below.

Major issues:

1. To control for differences in the rate of infection, staining for viral proteins should be added to all experiments. Interpretation of several results is limited without a control for infection.

We apologize for not including this control at the time due to the lack of a working antibody to detect SeV by WB. After testing various new antibodies from multiple companies, we have now identified a pan-SeV protein antibody that works in WB. We have reblotted key panels (for which we still had cell extracts) and found no difference in the infection rate between WT and PACT KO cells (**New Figure 2A, 2B, 2F, 4D, 4G, 7F, and Supplementary Figure 1A, 6H, 6I**). For the immunofluorescence experiment, we used FISH probes targeting the SeV genome to detect infected cells (see Question 2 for additional explanation). Again, we now showed no difference in infection rate between WT and PACT KO cells (**New supplementary Figure 1F-G and 2C-D**).

2. What is the relevance of quantifying number of foci per cell? As discussed by the authors and recently shown in the context of SeV infection (PMID: 37983241), SG are dynamic appearing/growing/disappearing at different times during infection. To better support the conclusions on figure 3 the authors should label infected cells in each condition and quantify

SG+ cells based on the number of infected cells.

We quantified the percentage of cells positive for SGs and the number of SGs by cells to reflect the highly dynamic nature of SG formation in the best way we could. We believe that providing supplementary information on whether a SG-positive cell has one or two SGs versus 10 to 15 SGs better illustrates the dynamic range of SG formation. Indeed, this suggests that the level of stress in cells with a higher number of SGs is more critical than in cells with fewer SGs. We used a similar strategy in our previous publications to characterize the level of SGs after viral infections, including in response to SeV infections (Manjunath et al. Nature Communications, 2023 / Oh et al. PNAS, 2024).

Following the reviewer's suggestion, we labeled infected cells using FISH probes targeting the SeV genome, developed by Carolina Lopez's lab (PMID: 28986577). We observed similar levels of infection between WT and PACT KO cells (**New Supplementary Figure 1F-G**). We then quantified the proportion of SG-positive cells and the number of SGs per cell, specifically in cells with a detectable FISH signal. Consistent with our previous data, SG levels in PACT KO cells were significantly higher than in WT cells (**New Supplementary Figure 2C-D**), further confirming our conclusion that PACT limits PKR-mediated stress granule formation in response to RNA viral infections. We would like to point out that we did not quantify stress granule formation only in the cells infected with SeV in **Figure 3B** and **Figure 3E** because we also wanted to highlight the formation of stress granules in uninfected cells in PACT KO cells (an important point for the second part of the manuscript). We thank the reviewer for suggesting this important control!

3. The main conclusion of the manuscript (Figures 6/7) is that cells can tolerate self-RNA due to PACT / ADAR1 interference with PKR activation. To support this conclusion, the authors need to directly demonstrate binding of PACT/ ADAR to endogenous RNAs and test if high levels of endogenous RNA overcome this blockage. Virus infection impacts several cellular processes. The proposed model is indeed interesting but must be directly validated.

These are critical points! An accompanying paper from Dr. Sun Hur's lab (Ahmad et al. BioRxiv: <https://doi.org/10.1101/2024.10.23.619951>) performed PACT fCLIP (formaldehyde-assisted RNA-protein crosslinking and immunoprecipitation) to analyze the binding of endogenous RNA to PACT. They found that PACT and PKR bind similar endogenous RNA, especially IR-Alu pairs and regions proximal to IR-Alus. Moreover, previous studies have shown that ADAR1 also binds similar types of RNA in cells (PMID: 25751603, PMID: 29395325). Together, these studies support that PACT and ADAR1 bind endogenous dsRNA in cells, strongly supporting our proposed model.

To determine whether high levels of dsRNA can overcome the inhibition of PKR activation mediated by PACT/ADAR1, we transfected cells with increasing doses of poly(I:C) and directly compared PKR activation to cells lacking both PACT and ADAR1. Transfection of a high amount of poly(I:C) stimulates PKR at similar levels to those of untransfected cells depleted for PACT and ADAR1 (**New supplementary Figure 6G**). In addition, PACT or ADAR1 overexpression was sufficient to suppress PKR activation after a high amount of poly(I:C) transfection (**New Figure 4H** and **New supplementary Figure 7A**), demonstrating that an increased level of PACT or ADAR1 can compensate for the increased levels of dsRNA in cells. Taken together, these new results reinforced our model that both PACT and ADAR1 are critical to protect cells against low levels of dsRNA in cells. We thank the reviewer for suggesting these important experiments.

4. A discussion of how a virus infection would impact the interaction of PACT /ADAR with host RNAs and the consequences for cell survival is needed.

This is a great suggestion! We have now added a discussion at the end of the manuscript on the impact of PACT/ADAR's interaction with host RNAs and its consequences for cell survival.

Minor issues:

1. Figure 1 SeV infection time does not match in results and legend (14 h vs 24h).

We apologize for the confusion. The CRISPR-Cas9 screen was performed 15 hours post-infection with SeV (14.5h of SeV infection + 30 min AHA labeling [SeV replication is still ongoing during the labeling]), while the following experiments were performed at 24 hpi. The reason we selected a shorter time for the CRISPR-Cas9 screen is to minimize the levels of PKR activation, allowing a higher enrichment of gRNA that enhances PKR in response to SeV in the FACS-sorted population 488 negative. Therefore, we selected the shorter time following SeV infection leading to PKR phosphorylation (**New Supplementary Figure 1A**).

2. Figure 1A: Switching the positions of 488+ and 488- to align with Figure 1B would help readers to understand the data easily.

We have now modified **Figure 1A** accordingly.

3. Line 129-130: "transduction" should be "infection", and I'd like to know which cell lines are being compared.

Sorry! We have now corrected the sentence and added the requested information on the cell line tested.

4. Figure 1C: The puromycin neg cells percentage in "KO PACT, SeV infection" is about 57.7% (15/26), which is much higher than the corresponding data shown in Figure 1D. What is the explanation for this?

Respectfully, we obtained a different percentage number. The immunofluorescence panel in Figure 1C shows 15 puromycin-negative cells and 26 puromycin-positive cells, corresponding to 36.58% of the cell population exhibiting translation arrest ($15/(15+26)*100$). This number is consistent with the data presented in the graphs in **Figure 1D**.

5. Figure 2: It seems clear that PACT inhibition of PKR activity is more effective in the context of poly I:C than SeV infection. A discussion on this observation should be included.

Respectfully, we are hesitant to make such a strong claim without extensive validation through a direct comparison of poly(I:C) and SeV. As shown in the **new Figure 4D and Supplementary Figures 1A, 6G**, the basal activation of PKR following SeV infection or poly(I:C) transfection is highly dependent on factors such as SeV MOI, poly(I:C) concentration, and the time point used

to assess PKR phosphorylation. Therefore, to assert that PACT inhibition of PKR activity is more effective in the context of poly(I:C) transfection compared to SeV infection, a comprehensive side-by-side comparison under multiple conditions (e.g., time, MOI) would be necessary.

6. The PACT complementation was done by lentivirus transduction, which should be better described. It was difficult to figure out the method from the current description (some people complement by transfection).

We apologize for the lack of clarity in our methods section. To address this, we have expanded it to provide a more detailed explanation of the cell complementation approach used in this study with a lentiviral system. All experiments involving the overexpression of PACT-WT, PACT-EA, and ADAR1-p150 were conducted using stable cell lines generated through lentiviral transduction of the corresponding constructs.

7. Figure 4D: a qPCR data for SeV genome replication is necessary to show SeV infection of different MOI.

We reblotted the cell extract used for this panel with an antibody that detects SeV proteins. We now show a comparable SeV infection rate across all the samples (**New Figure 4D**).

8. Figure 4D and E use overexpression of PACT with different levels of viral dsRNA. Since the question is whether different expression of PACT would influence PKR activation by viral RNA, I'd like to see same amount of dsRNA (same MOI of SeV infection) but different amount of PACT plasmids transfection in PACT KO cells to validate the conclusions. SeV Cantell contain defective interfering particles that will reduce viral replication and the amount of dsRNA available, therefore the experiment is not conclusive as presented.

This is an important point! We have now performed the experiment suggested by the reviewer by generating two stable cell lines expressing homogeneous but different levels of PACT using two distinct lentiviral vector systems: pLenti (high PACT expression) and pInducer20 (low PACT expression). While high PACT expression completely suppresses PKR activation following SeV infection, lower expression of PACT using the same SeV MOI leads to only a partial suppression of PKR activity (**New Supplementary Figure 6I**). These findings demonstrate that differential PACT expression results in a dose-dependent response of PKR activation.

Additionally, we also transfected the cells with poly(I:C) to rule out the role of defective interfering particles affecting the amount of dsRNA available during viral infection. Similar to the result obtained after SeV infection, we observed decreased PKR activation with increased PACT expression after poly(I:C) transfection (**New Figure 4H and New Supplementary Figure 6I**). Taken together, these results strengthen our model that PACT prevents viral dsRNAs to induce PKR.

9. Discuss whether acute and persistent infections could be related to this, considering the differences in dsRNA levels between them.

The reviewer raises an important point. Given the proposed role of defective interfering particles in establishing persistent viral infections, it is intriguing to consider whether different levels of dsRNA in SeV-infected cells influence the outcome—acute versus persistent infection. However, since this manuscript primarily focuses on the initial cellular response to RNA viral infection, an extended discussion on the role and consequences of DVGs in SeV-infected cells, particularly in the context of acute versus persistent infection, may be beyond the scope of this study (We did not mention DVG biology in this manuscript; therefore, introducing this concept would require an extended explanation). Additionally, without supporting experimental evidence, such a discussion could be seen as too speculative and potentially divert the reader's attention from the manuscript's primary focus. If the reviewer agrees, we suggest this topic would be better suited for a follow-up study.

Line 31: should read identified

Line 32: deficient on...

Line 255: to test this hypothesis...

In general needs editing

Thanks! We have now corrected these typos.

What is synthetic lethality?

Synthetic lethality occurs when the disruption of either of two genes individually does not affect cell viability, but the simultaneous disruption of both genes leads to a loss of viability. In other words, the two genes compensate for each other's function, and the combined loss of their activity creates a lethal outcome. This is precisely what we observed with PACT and ADAR1. The absence of either PACT or ADAR1 alone did not significantly affect cell survival; however, the simultaneous absence of both genes resulted in cell death.

Reviewer #3 (Remarks to the Author):

This study investigates cellular mechanisms that regulate activity of the innate immune sensor protein kinase R (PKR). They address important fundamental questions about how PKR is regulated to distinguish self double-stranded RND (dsRNA) from viral RNA. They describe new functions for the protein PACT, suggesting it might interfere with PKR activation in response to viral infection. They further demonstrate that PACT cooperates with ADAR1 to suppress PKR activation from self-dsRNAs in uninfected cells. They propose that PACT and ADAR1 together act as a barrier against PKR activation to generate a threshold for levels of endogenous dsRNA that can be tolerated in cells. In elegant and rigorous experiments they first identify PACT through a CRISPR screen for suppressors of translation arrest during RNA viral infection. They then show that PACT inhibits PKR phosphorylation during RNA viral infection by immunoblotting in knockout cells. They demonstrate that PACT can suppress PKR-mediated arrest of protein translation and formation of stress granules in response to virus infections. Structural modeling is used to investigate interaction with dsRNA, and they suggest that higher PACT expression can inhibit PKR activation by protecting the dsRNA. A second CRISPR screen in PACT knockout cells identifies ADAR1 as synthetically lethal, and they suggest they act synergistically to suppress PKR activation. There are just a few key experiments that could be done to complement findings and further support conclusions. Overall, the study makes interesting observations that offer new perspectives to understand the activation of PKR in response to

self-RNAs and how these dsRNA-binding proteins participate in the modulation of the antiviral response regulated by PKR. Experiments are well executed and described. It is a significant contribution that will be appreciated by those working on innate immune sensors and dsRNA binding proteins.

We thank the reviewer for appreciating our work and recognizing the novelty of our study! We have now addressed the reviewer's comments and suggestions.

Specific Comments:

1) In Figure 1 they employ an elegant technique called CRISPR-Translate to screen a library to identify factors that regulate PKR and host translation shutoff. This identified PACT. The authors could further explore and describe the other hits that past their threshold that fall between PACT and ADAR1. The translational inhibition observed in Figures 1C-1E) show that it is detected in only a small proportion (around 10%) of SeV-infected cells at 24hpi. Why is this the case and does this suggest other proteins are also involved?

We are currently testing and validating the other hits identified using our CRISPR-Translate screening method shown in **Figure 1A-B**. However, testing all of them and characterizing the mechanisms regulating translation in response to SeV—some of which may be entirely unrelated to PACT or ADAR1—will take many months or even years. Therefore, we hope the reviewer agrees that it would be more suitable to use these results for a future manuscript.

Yes, the reviewer is correct! The initial observation that only a low percentage of SeV-infected cells trigger translation arrest suggested that other proteins are involved in preventing translation shutdown in response to SeV infection. This motivated us to perform a CRISPR-Cas9 screen to identify them. Both PACT and ADAR1 are essential in protecting SeV-infected cells from triggering translation arrest, but additional factors, yet to be identified, may also play a role.

2) They suggest that PACT inhibits PKR primarily through its RNA-binding activity. Although data linking PACT overexpression to inhibition of PKR phosphorylation are compelling, further evidence is necessary to support that PACT interferes with PKR binding to dsRNA. However, they do not experimentally demonstrate that overexpression of PACT prevents PKR binding to dsRNA. Additional experiments could be done to support comments in lines 274 and 275 by demonstrating competition in vitro or Co-IP experiments using dsRNA antibodies (J2 or 9D5) testing whether PKR pulldown is decreased in cells overexpression PACT. In addition, pl:C transfection was shown to induce cytoplasmic dsRNA-induced foci (dRIFs) where PKR, PACT and ADAR1 accumulate (PMCID: PMC9388085). The authors could explore how the overexpression of PACT (and ADAR1) affects the recruitment of PKR to these sites of dsRNA accumulation. In addition, if it is really through a threshold model, they should be able to generate inducible systems to complement the knockout and demonstrate a dose response to effects on PKR activation.

These are important points! An accompanying paper from Dr. Sun Hur's lab (Ahmad et al. BioRxiv: <https://doi.org/10.1101/2024.10.23.619951>) focused on characterizing in detail using in vitro and biophysics systems shows how PACT impacts the dsRNA binding with PKR. They elegantly revealed that the inhibition of PKR by PACT is not necessary by competition through the sequestration of dsRNA by PACT. Indeed, they showed that PACT forms a barrier on dsRNA, restricting PKR's movement and thereby reducing the likelihood of PKR molecular collisions, dimerization, and activation. Moreover, we now show the colocalization of PACT and

PKR at dRIF in cells (similar to the results obtained from Dr. Parker's group Corbet et al. PNAS 2022) (**New Supplementary Figures 7B-C**), further supporting the model from Dr. Hur's lab where both PKR and PACT bind the same dsRNA, but PACT prevents the activation of PKR likely by preventing PKR dimerization of the dsRNA.

In addition, we performed the experiment suggested by the reviewer to determine how PACT or ADAR1 overexpression impacts PKR dRIF formation after poly(I:C) transfection. We found that overexpression of either PACT or ADAR1 significantly reduces PKR dRIF formation in poly(I:C)-transfected cells (**New Supplementary Figure 7D**). This suggests that when PACT or ADAR1 is overexpressed, high levels of binding to dsRNA may prevent or compete with PKR for dsRNA binding. Therefore, we have incorporated both models into our interpretation of PKR suppression by PACT. *“Taken together, we propose that PACT prevents hyperactivation of PKR, either through direct competition or by creating a barrier that prevents the assembly of two PKR monomers bound on the dsRNA (Figure 4J)”*

Finally, to address the reviewer's question about whether differential expression of PACT results in a dose-dependent response of PKR activation, we established two stable cell lines expressing homogeneously PACT but different levels of PACT using two distinct lentiviral vector systems: pLenti (high PACT expression) and pInducer20 (low PACT expression). While high PACT expression completely suppresses PKR activation following SeV infection or ADAR1 depletion, lower expression leads to partial suppression (**New Supplementary Figure 6I**). These findings demonstrate that differential PACT expression results in a dose-dependent response of PKR activation.

3) If ADAR1 and PACT have redundant roles (as argued in lines 325-326), it should be shown by rescue in the phosphorylation of PKR in PACT KO cells following overexpression of ADAR1. It is shown that ADAR1 or p150-ADAR1 are sufficient to prevent PKR activation by endogenous RNAs. This should be demonstrated in PACT KO cells during SeV infection or pI:C transfection. This should be done by overexpressing ADAR1, p150-ADAR1 or catalytic inactive mutant (ADAR1-p150-E912A).

This is a great suggestion! We have now performed the experiments suggested by the reviewer by generating stable PACT KO cell lines overexpressing ADAR1-p150. We then infected PACT KO cells with SeV, either overexpressing or not overexpressing ADAR1-p150, and monitored PKR activation. Our results show that PACT KO cells exhibit strong suppression of PKR phosphorylation levels when ADAR1-p150 is overexpressed (**New Figure 7F**). We obtained similar results with PACT KO cells transfected with poly(I:C) (**New Supplementary Figure 7A**). Together, these new results further confirm our model and the redundant roles of ADAR1 and PACT in maintaining a tolerable threshold level of dsRNA in cells. We thank the reviewer for suggesting these important experiments.

Minor Issues

1) Statistics: In all figures where statistic tests were performed, two-tailed t-test were used. In this approach looking for decrease in translation as an example, when comparing two groups, one-tailed tests should be applied. Authors however also compare different conditions. For these, analysis of variance (ANOVA) should be applied rather than t-tests for selected conditions in the same dataset.

Thanks for suggesting better statistical tests for our panels. We have now recalculated all the p-values using the correct statistical test.

2) Line 129. They use the word “transduction” when instead they are referring to infection.

Thanks for catching this mistake! We have now corrected the sentence.

3) Line 128-132. Authors picked 14h post infection with SeV as the best time point to screen for AHA positive and negative cells. Authors could provide a time course showing PKR phosphorylation over time in U2OS cells infected with SeV as well as a control for showing infected cells using a viral protein as marker.

We now show the time-course experiment we initially conducted, demonstrating that PKR activation begins 15 hours post-infection. In addition, we included a WB confirming the presence of viral protein in the samples infected with SeV (**New Supplementary Figure 1A**).

4) Line 193 (Figure 2D and 2E). Both control viral proteins are marked as VP1 in W B panels. Is this the correct designation?

Yes, this is the correct designation. Both PV and EV-A71 express a protein named VP1. However, two different antibodies specific for each virus were used for these panels, as indicated in **Supplementary Data 2**.

5) Line 216. PKR structural modeling is not explored in this section but is mentioned in the title.

The reviewer is correct. We have now removed PKR from the title of this section.

6) Line 334 (Supplementary Fig 6F). Short and long exposure panels are inverted for ADAR1.

Thanks for catching this mistake! We have now corrected this panel.

7) Line 357-374. Discussion Paragraph 1 lacks references.

Apologies for omitting references in this paragraph. We have now added several references accordingly.

8) Line 370. Using the term “cellular sensors” imply PACT and ADAR1 are sensing dsRNAs which will lead to a downstream response. In the paper it is argued that these proteins rather bind to dsRNA to prevent PKR activation in response to self-RNAs or its overactivation during viral infections.

The reviewer is correct. We have now removed the term “cellular sensor” and modified our statement.

REVIEWER COMMENTS

Reviewer #1 (Remarks to the Author):

The authors have addressed all of my comments and I am supportive of publication.

Thanks for your support!

Reviewer #2 (Remarks to the Author):

We thank the authors for a wonderful job in addressing the concerns.

We appreciate your kind words about our work!

Reviewer #3 (Remarks to the Author):

The authors have done a commendable job in addressing all our prior critiques and responding to all reviewer comments. This work makes a nice contribution to the field.

Thank you for your appreciation of our work!